# Birds multiplex spectral and temporal visual information via retinal On- and Off-channels

**Marvin Seifert[1] ✉, Paul A. Roberts [1], George Kafetzis[1], Daniel Osorio [1] ✉ & Tom Baden [1,2] ✉**

In vertebrate vision, early retinal circuits divide incoming visual information into functionally opposite elementary signals: On and Off, transient and sustained, chromatic and achromatic. Together these signals can yield an efficient representation of the scene for transmission to the brain via the optic nerve. However, this long-standing interpretation of retinal function is based on mammals, and it is unclear whether this functional arrangement is common to all vertebrates. Here we show that male poultry chicks use a fundamentally different strategy to communicate information from the eye to the brain. Rather than using functionally opposite pairs of retinal output channels, chicks encode the polarity, timing, and spectral composition of visual stimuli in a highly correlated manner: fast achromatic information is encoded by Off-circuits, and slow chromatic information overwhelmingly by On-circuits. Moreover, most retinal output channels combine On- and Off-circuits to simultaneously encode, or multiplex, both achromatic and chromatic information. Our results from birds conform to evidence from fish, amphibians, and reptiles which retain the full ancestral complement of four spectral types of cone photoreceptors.

Vertebrate retinal circuits process a stream of spatial, temporal, and spectral information into parallel channels for transmission to the brain[1]. The number of channels, what each channel encodes, and why, is a subject of active debate[2], but some general principles have emerged[3]. For example, that to save energy, and to expand dynamic range, retinal circuits divide the visual signal into an approximately equal number of On- and Off-channels[4], and into transient and sustained channels[5–7]. Information about polarity and kinetics is thereby represented by independent 'elementary building blocks' with decorrelated signals[5]. Similarly, wavelength information is efficiently funnelled into a subset of chromatic channels in parallel to the achromatic channels that dominate the retinal output[8,9].

These long-standing principles are overwhelmingly based on research on mammals, such as mice, primates, cats, and rabbits, and evidence from non-mammalian species is beginning to question their generality. For example, zebrafish lack the approximately balanced representations of On- and Off-, and of transient and sustained channels of mammalian retinas[10–16]. Similarly, the profusion of colour-opponent responses in turtles[17], or the heavy dominance of OnOff channels in salamanders[18,19], do not easily map onto the current framework. Yet, a systematic overview of retinal computations is lacking for most non-mammalian lineages, and especially for birds[20,21]. This is in part due to the difficulty of recording from the ex-vivo bird retina[22–24], where minor mechanical insult causes pathological waves of depression[25]. Consequently, to date insights into the operation of avian retinas are largely[22,26] restricted to non-invasive techniques such as electroretinograms[27,28].

Working on the retina of poultry chicks[20] (*Gallus gallus*, Fig. 1a–c) we have overcome these limitations by suppressing pathological signal spread during dissection (Supplemental Fig. S1). We can record light-driven spiking activity from neurons in the retina's ganglion cell layer with a high-density multielectrode array (MEA) for several hours (Fig. 1d, e), and now present what to our knowledge is the first large-scale study of light-driven retinal function in a bird.

---

[1]School of Life Sciences, University of Sussex, Brighton, UK. [2]Institute of Ophthalmic Research, University of Tübingen, Tübingen, Germany. ✉e-mail: m.seifert@sussex.ac.uk; d.osorio@sussex.ac.uk; t.baden@sussex.ac.uk

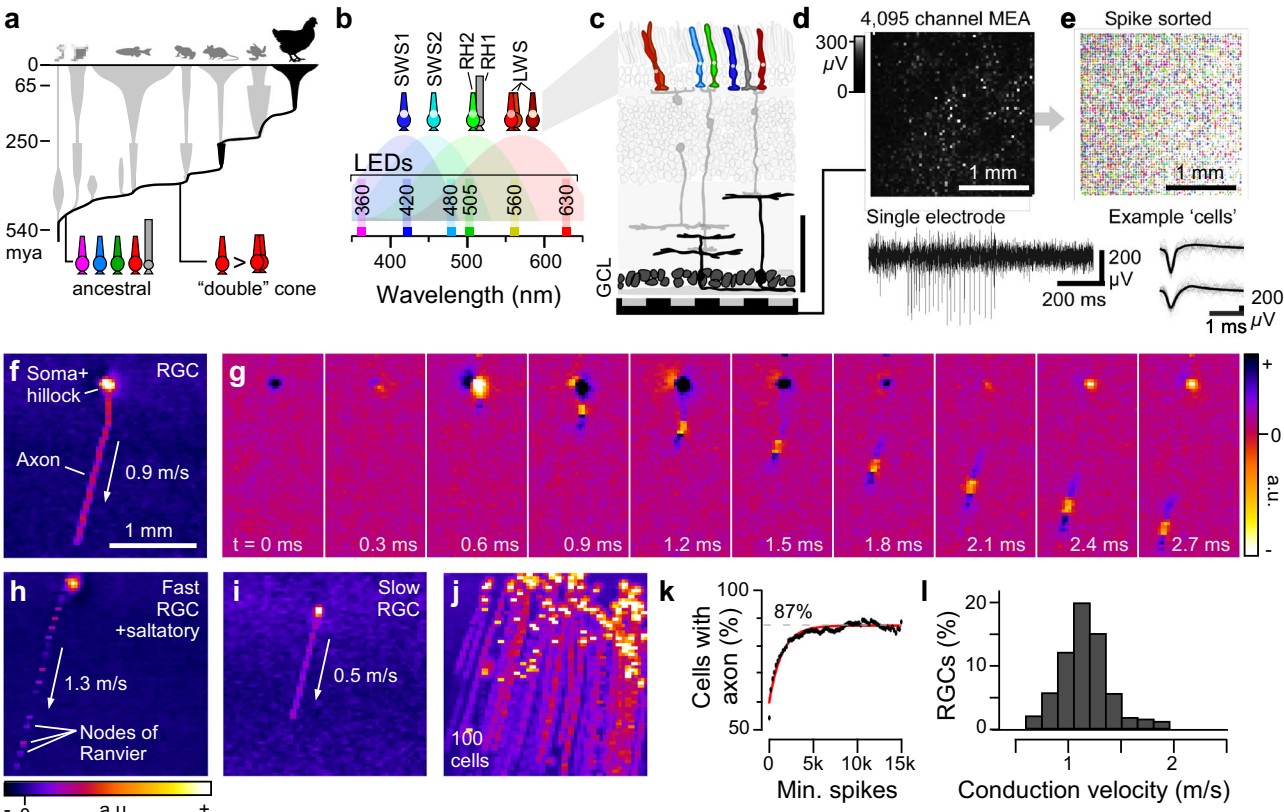

**Fig. 1 | Recording light-driven spiking activity from the chick retina.**
**a** Phylogenetic tree of extant vertebrates based on ref. 30, indicating key evolutionary transitions in photoreceptor complements leading to birds. **b** Summary of birds' seven types of ciliary photoreceptors and their expressed opsins (top), the respective spectral sensitivity functions of their four cone opsins (shadings), and spectral positions of the six LEDs used for visual stimulation. **c** Schematic of chicken retina modified from ref. 2 illustrating the multielectrode array (MEA) recoding strategy. GCL, ganglion cell layer. The coloured cells in the top are the photoreceptors, while the dark grey-shaded neurons in the bottom depict retinal ganglion cells (RGCs, the retina's output neurons), potentially alongside local interneurons called displaced amacrine cells (dACs). **d** Representative recording frame from the 64 × 64 MEA array, indicating localised activity on a subset of electrodes (top) and example trace from a single electrode (bottom). **e** Overview of all spike-sorted units detected in an example recording (top) and spike waveforms of two example 'cells' (bottom). **f** Minimum intensity projection across 50 time bins (corresponding to ~3 ms) of a representative electrical image (EI) computed from a single spike sorted unit, revealing the position of the soma, hillock and axon, as indicated. **g** Time-series from the EI shown in (**g**), illustrating how the spike travels down the axon towards the optic disc. **h**–**j** As (**f**) but for two different spike sorted units (**h**, **i**) and for 100 such units superimposed (**j**). Note that the axonal footprint in (**h**) is fractionated, indicating saltatory conduction, while the footprints in (**f**) and (**i**) are continuous, indicating non-saltatory conduction (cf. ref. 23). **k** Fraction of EIs with a clearly detectable axon (y-axis, Methods) plotted for different subsets of the full EI dataset, staggered by the minimum number of spikes available to compute each EI (x-axis). The asymptote of an exponential fit to this data (red) indicates the presumed fraction of EIs that have an axon independent of the signal quality was ~87%. **l** Distribution of axonal conduction velocities estimated from all presumed RGCs (n = 842). Source data are provided as a Source Data file.

Like most birds, the chicken eye has seven[29] distinct types of photoreceptors (Fig. 1a–c). These include rods (RH1), the full complement of four ancestral[30,31] single cones (SWS1, SWS2, RH2, LWS, Fig. 1a), and the principal and accessory members of the double cone (expressing LWS opsin). Within the outer plexiform layer's three strata[20,32], these photoreceptors differentially feed into more than 20 types of bipolar cells[32,33], which in turn drive more than 40 types[20,33] of retinal ganglion cells (RGCs), the retina's output neurons.

We find that unlike in mammals[2,34], the retinal ganglion cells of chicken represent information about the polarity, kinetics, and wavelength of light stimuli in a highly correlated manner, with fast and achromatic Off-circuits, and slow and chromatic On-circuits. Moreover, most retinal outputs combine both On- and Off-circuits to simultaneously inform about both sets of information.

## Results

### Most recorded cells correspond to retinal ganglion cells
We recorded from n = 17 pieces (~2–3 mm in diameter) of dorsal retina (N = 14 male poultry chicks), yielding n = 3987 spike sorted cells that passed a minimum response criterion (Methods, cf. Supplemental

Fig. S2). To estimate what fraction of these cells stemmed from retinal ganglion cells (RGCs), which are the retina's output neurons, rather than from local interneurons such as displaced amacrine cells (dACs)[3], we computed the 'electrical image' (EI)[23] for a subset of spike-sorted cells (Fig. 1f–l, Methods, see also Supplemental Video S1). This image permits inferring (i) each cell's spike initiation zone and soma, (ii) the trajectory of its axon, if present and well-attached (e.g. Fig. 1f), (iii) its conduction velocity (Fig. 1g), and (iv) whether it displayed saltatory (Fig. 1h) or non-saltatory spike propagation (Fig. 1f, I, cf. Fig. 1j). We reasoned that axon bearing cells were likely to be RGCs. In contrast, axon-less cells likely included dACs and any RGCs whose axons were not detected (e.g. because they were on the array-edge, or because their axon was far from the retinal surface). No EIs had multiple axons radiating from the centre, as for polyaxonal dACs in mammals[35]. Accordingly, we took the fraction of EIs with a clear axon (Methods) as the lower bound of RGCs. The reliability of axon-detection scaled with the number of available spikes before reaching an asymptote at ~87% (Fig. 1k), suggesting that at least 87% of recorded cells were RGCs. For these, we then quantified conduction velocities (Methods), which in line with recent work[23] varied by a factor of more than four, from

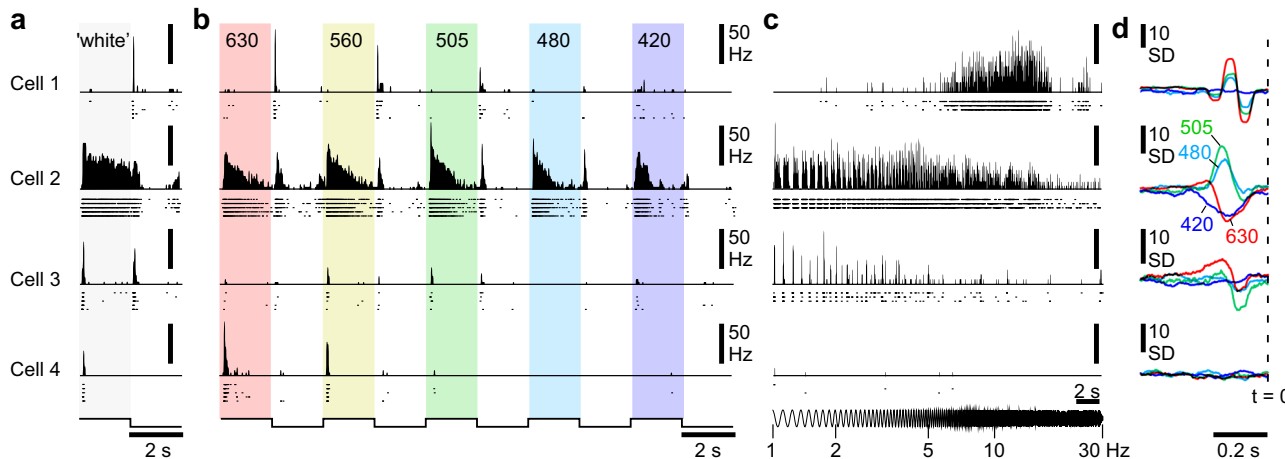

**Fig. 2 | Example light responses. a–d** Spiking light responses of four example cells to the battery of presented stimuli, including the 100% contrast 'white' (**a**) and 'coloured' steps (**b**) with centre wavelengths in nm indicated in the top, the chirp stimulus (**c**) and, correspondingly with (**b**), the groups of spectral kernels recovered from spectral noise stimulation (**d**). Solid histograms indicate the trial-averaged means per cell, with spike-rasters beneath showing the individual repeats. Note that in (**b**) the 6th colour step response (near-UV) is not shown as cells tended to not respond at this wavelength. The full responses are included in the online plotter (see below). Source data are provided as a Source Data file.

<0.5 m/s for the slowest non-myelinated axons, to >2 m/s for the fastest axons (Fig. 1l).

### A large-scale functional survey of the avian retina

To begin understanding how the chick retina functions, we focussed on temporal and colour vision. For this, a set of six spectral LEDs, from 360 (UV) to 630 nm (red) were chosen to cover the chicken's full spectral range. Following strategies originally established for work with fish[10,36], these LEDs were calibrated to 20 (UV) – 100 (red) nW to approximate the "ramped" spectral composition of daylight[37] in the chicken's natural habitat (Methods).

We presented four complementary sets of widefield light-stimuli, which were chosen to characterise achromatic and spectral responses within a limited recording time, and allow direct comparison with other species[10,38–40] (Methods).

(i) A series of 'white steps' (WS, i.e. all LEDs concurrently activated) from dark to light at different contrasts (2 s On, 2 s Off at 100-10% contrast; Fig. 2a showing 100% step).

(ii) A corresponding series of 100% contrast 'coloured steps' (CS, i.e., each LED individually activated at full power) from dark at different wavelengths. (630, 560, 505, 480, 420, 360 nm, Fig. 2b showing the first five, cf. Fig. 1b).

(iii) A 100% contrast 'white' chirp stimulus, exponentially accelerating from 1–30 Hz over 30 s (Fig. 2c).

(iv) A 20 min 20 Hz pseudorandom sequence of 'coloured' noise (420, 480, 505, 630 nm) to compute spectral kernels, which correspond to the linear component of the impulse response[41] (Fig. 2d).

Because our multielectrode array is opaque, stimuli were presented from the photoreceptor side, by-passing the spectral filtering by oil droplets that occurs in the intact eye[42].

Figure 2 shows responses of four individual cells to the stimuli, chosen to capture some of the diversity found. Cell 1 responded transiently to light offset (Fig. 2a), preferred long- over short-wavelengths (Fig. 2b), and had bandpass temporal tuning centred around 10 Hz (Fig. 2c). The cell's spectral kernels were correspondingly narrow, indicating high-frequency temporal tuning, and confirmed the cell's non-opponent long wavelength preference (Fig. 2d). Cells 2 and 3 were both OnOff cells but differed in their kinetics and spectral tuning. Cell 2 was On-sustained but Off-transient, with broad, low-pass tuning to temporal flicker, and its spectral kernels were colour opponent. Cell 3 had transient responses to light onset and offset with a preference for 'white' over 'coloured' light (compare Fig. 2a, b) and low-pass tuning to temporal flicker that did not respond above 5 Hz. Cell 4 responded transiently to light onset with little response to temporal flicker or to spectral noise (Fig. 2a, c). Instead, it had selective spectral sensitivity to 'red'- and 'yellow' wavelengths, alongside a relatively weak response to 'white' light (Fig. 2b).

### An interactive online data explorer of avian retinal ganglion cell functions

To identify major functional structures across the dataset we clustered responses to the four stimuli using a Mixture of Gaussian model, which assigns cells to clusters (Gaussian distributions) based on expectation maximisation (Methods). This was followed by manual curation (Methods), yielding $n = 36$ clusters, of which $n = 22$ had a minimum of 20 members and were retained for further analysis ($n = 3914$ cells, 98.1%). Figure 3a plots a representative selection of these clusters, while all 22 clusters are shown in Supplemental Fig. S3. The full dataset, including further analysis, is interactively plotted and available for download at http://chicken-data.retinal-functomics.net/.

To what extent the 22 clusters correspond to anatomically and molecularly distinct types of RGCs and/or dACs is unknown. However, the 41 transcriptomically described RGC types in the adult chicken[33] and a similar anatomical RGC diversity[20] imply that some of our clusters comprise more than one type. Further probing the response space may well reveal additional functional diversity. Similarly, inherent to extracellularly recording the activity of many densely packed neurons in parallel, our MEA dataset likely comprises a sampling bias. Considering the high density of neurons in the GCL of chicks[20], we estimate that on average our MEA and spike sorting approach (Methods) yielded signals from ~10–20% of all RGCs on the array (out of ~30,000), but subsequent filtering by a series of conservative quality criteria (Methods) further reduced this number to about 1% (~300 well-isolated units, cf. Supplemental Fig. S4e). Consequently, the reported proportions of functional types may not accurately reflect their proportions in the animal, and some functional signatures might be missed altogether. In the absence of complementary data obtained by different recording techniques (e.g. by 2-photon imaging, or single electrode sampling), it is currently not possible to assess the size or nature of this bias.

We now describe the basic complement of neural responses in the chick retina. For simplicity, and unless otherwise specified, analysis

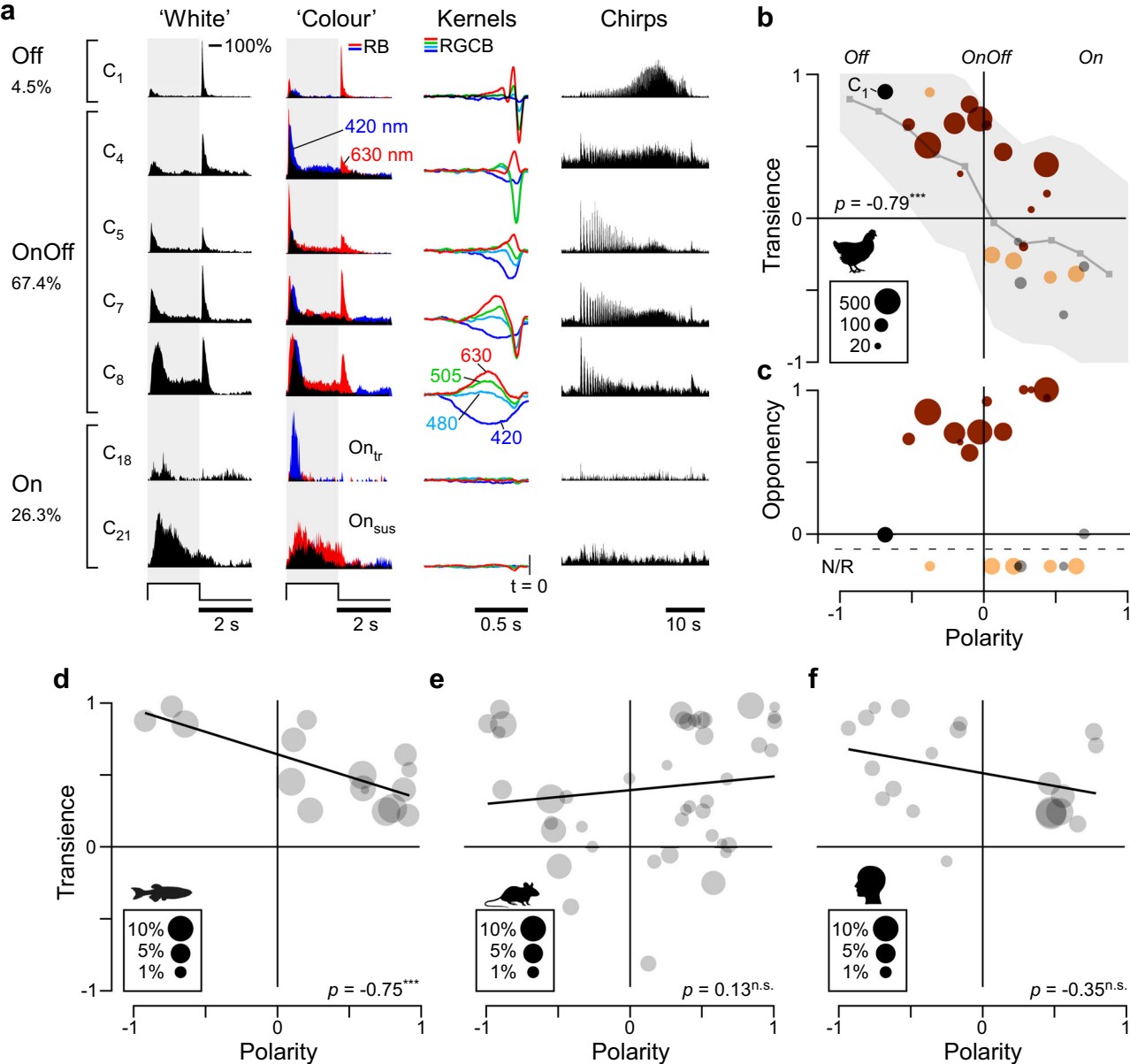

**Fig. 3 | A tight link between polarity, kinetics, and colour opponency.**
**a** Overview of selected cluster means to the full set of presented stimuli, illustrating some of their systematic differences. For simplicity, only the 100% contrast response is shown for WS, while only the red- (630 nm) and blue-responses (420 nm) are shown for CS. The full set of cluster means are shown in Supplemental Fig. S3. The entire dataset alongside a basic, cluster wise analysis, is interactively plotted and available for download at http://chicken-data.retinal-functomics.net/. **b**, **c** Relationships of transience (**b**) and colour opponency (**c**) with polarity for all clusters based on their mean 100% contrast WS responses (Methods). Symbol size denotes the number of cells allocated to a cluster as indicated, while colouration indicates the four response groups: Off (black), OnOff (brown), On (orange/grey). Note that On-responses are further divided by the kinetics of their CS-responses

into a transient (orange) and a sustained group (grey)−cf. Supplemental Fig. 4a, b. **b** Further shows the mean ± SD shadings for the same relationship based on all individual cells (light grey). A positive transience index denotes a transient cell, 0 sustained, and negative a temporally increasing response, as evaluated by comparing the peak response in two time-windows following the step transition: 80–160 ms and 240–2000 ms (Methods). **d**–**f** As (**b**), transience and polarity indices of RGCs found in the retina of larval zebrafish (**d**, based on ref. 10), mice (**e**, based on ref. 43), and humans (**f**, based on ref. 44). Linear correlation tests, two-sided: Chicken: $p < 0.001$; Zebrafish: $p < 0.001$; Mouse: 0.41; Human: 0.13. Note that in (**d**–**f**), the colour code from (**b**) does not apply. For further detail see also Supplemental Fig. S4d–g. Source data are provided as a Source Data file. Human silhouette in (**f**) from silhouettegarden.com.

was performed on the cluster means. Cell-wise summaries of key indices are available in the interactive online data explorer.

**Response polarity, kinetics, and colour opponency are linked**
We began by exploring the clusters' light response properties based on the polarity (On, Off, OnOff) and kinetics ('transient' or 'sustained') of their 'white' step responses (Fig. 3b, see also Supplemental Fig. S4a, b, Methods). This revealed that cells in most clusters responded both to

light onset and offset, rather than forming well-segregated On- and Off-channels. Second, polarity and kinetics were linked: Off-dominated clusters were transient, while On-dominated clusters were generally more sustained. Third, analysis of the spectral kernels (Methods) revealed that colour processing was also linked to polarity: Most OnOff clusters had high-amplitude colour opponent kernels, whereas most Off- or On-dominated clusters were non-opponent or had low-amplitude kernels (Fig. 3c). As subset of these latter clusters

responded more strongly to 'coloured' steps than to the white steps that were used to define their polarity and transience (discussed below) we also computed these measures from their peak responses to the 'coloured' stimuli (Supplemental Fig. S4a, b). This allowed characterisation of On-dominated clusters as distinct transient and sustained groups.

Accordingly, our 22 clusters fell into four groups:
(i)   Off, transient, non-opponent ($C_1$, $n = 180$ cells, 4.5%)
(ii)  OnOff, kinetically mixed, colour-opponent ($C_{2-13}$, $n = 2688$ cells, 67.4%)
(iii) On, transient, non-opponent ($C_{14-19}$, $n = 784$ cells, 19.7%)
(iv)  On, sustained, non-opponent ($C_{19-22}$, $n = 262$ cells, 6.6%)

The cluster-means shown in Fig. 3a illustrate these groups, and their key differences (Supplemental Fig. S4c summarises how clusters and response groups are distributed across experimental sessions). The groups and their associated analysis are colour-coded in black, brown, orange, and grey, respectively (e.g. in Fig. 3b, c).

## OnOff dominance as a hallmark of non-mammalian retinas?
To compare chick RGC responses to those of other vertebrates, we sourced widefield 'white-step' response datasets of RGCs from larval zebrafish[10], mice[43], and humans[44] and computed their corresponding distributions of polarity and transience (Fig. 3d–f, cf. Supplemental Fig. S4d–g). This revealed similarities between the chick and zebrafish datasets, which systematically differed from the mammalian systems.

Chicks (Fig. 3a), and to a lesser extent zebrafish (Fig. 3d), feature a sizable complement of OnOff channels (~65–70% and 30–35%, respectively; cf. Supplemental Fig. S4d, f, g). OnOff channels are also dominant in salamanders[18] and turtles[45]. In contrast, segregated On- and Off-channels predominate in mice and humans (Fig. 3e, f). Even the few well-described mammalian OnOff channels, such as the OnOff direction selective RGCs of mice or the small bistratified RGCs of primates, are dominated by either their On- or their Off-components when probed with 'white' steps of light. Second, the link between polarity and kinetics observed for chicks (Fig. 3a) was also a feature of the zebrafish dataset (Fig. 3d), but no such trend was detectable for mice or humans, whose RGCs occupy the full coding space encompassed by polarity and kinetics (Fig. 3e, f).

What might be the benefit for chicks, and perhaps also of other non-mammalian vertebrates, to have combined OnOff-channels, rather than segregated channels for encoding On- and Off-events? And why are polarity, kinetics, and colour opponency linked?

To address these questions, we now analyse each system in turn: Off-, On- and then OnOff. We find that Off-channels encode fast achromatic contrast, On-channels primarily deal with spectral information, while their combination into OnOff-channels simultaneously carries information about both achromatic and chromatic aspects of the visual stimulus.

## Off-circuits encode achromatic temporal contrast
No cluster yielded exclusively Off-responses, but the most Off-dominated cluster $C_1$ was distinctive as the only cluster having large amplitude kernels without appreciable colour opponency (Fig. 4a, cf. Fig. 3b, Supplemental Fig. S3), and the fastest response, with centroids (Methods) peaking above 5 Hz for red-, green- and cyan- components (Fig. 4b, c). Correspondingly, the cluster's chirp response (Fig. 4d, top) exhibited bandpass tuning that peaked around 10 Hz (Fig. 4e, f), and was phase locked to temporal flicker up to the maximum tested frequency of 30 Hz (Fig. 4g, h). Most other clusters had low pass tuning to the chirp stimulus and did not follow flicker above 5 Hz. The exception was $C_2$, the second-most Off-biased transient cluster (Fig. 3b), which was allocated to the OnOff group due to its strong colour opponency (Fig. 3c). $C_2$ frequency tuning matched $C_1$, as described by their spectral kernels (Fig. 4b, c), and for temporal flicker (Fig. 4d–g), but had a

weaker preference for high- (4– 14 Hz) over low- (<2 Hz) frequencies (Supplemental Fig. 5a, c, d cf. Fig. 4e) and weaker phase locking (Fig. 4g). $C_{1,2}$ both had smaller On- than Off- responses to intensity steps.

Although $C_{1,2}$ were the two most Off-dominated clusters and had the fastest temporal tuning, their flicker responses were On- at low and Off- at high-frequencies (Fig. 4h). Importantly the same reversal of response polarity applied to all clusters with any phase locking (Supplemental Fig. 5b). Thus, it appears that chicks use On- and Off-circuits to encode slow and fast temporal contrast respectively. Axonal conduction velocities of the four groups supported this conclusion: Off-RGCs were faster than OnOff-RGCs, which in turn were faster than On-RGCs (Fig. 4i).

Beyond speed, Off-circuits had three properties suited to encoding achromatic intensity, exemplified by cluster $C_1$. First, Off-responses had linear contrast-response functions to white steps (Fig. 4j). Second, their spectral tuning matched a log-sensitivity function of the LWS opsin (Fig. 4k), which is expressed in red single cones and the double cone (Fig. 1b)—the latter having long been implicated in fast, achromatic vision[46]. Third, key features of Off-responses were common to all clusters, namely their kinetics to white-steps (Supplemental Fig. S4d), their contrast-response functions (Fig. 4l), and their spectral tunings (Fig. 4m). By comparison On-responses were diverse (cf. Fig. 5, discussed below).

Cluster $C_1$ and $C_2$, which contain the chick's rapid achromatic Off-circuits, comprised 4.6% and 2.8% of recorded cells respectively. By comparison, the functionally similar primate parasol cells comprise 10–16% of RGCs[47], while the types of alpha cells in mice make up about 5%[38,48–50]. Unlike in chicks, these mammalian fast achromatic contrast systems have equal proportions of On- and Off-cells.

## On-circuits encode wavelength information
In contrast to the homogeneous, fast, and achromatic Off-circuits, On-circuits tended to be heterogeneous, slow, and spectrally nuanced. For example, $On_{tr}$-cluster $C_{18}$ had a highly selective On-response to blue-light (Fig. 5a–c), while cluster $C_{15}$ was selective for red-light (Fig. 5d–f). Both clusters were selective for coloured over white light with the peak response to the spectral stimuli exceeding the corresponding 'white' response. Both $C_{15}$ and $C_{18}$ lacked any overt sign of colour-opponency but their' spectral tuning was narrower than those of the spectrally closest opsins (Fig. 5b, e), unlike the Off-cluster $C_1$ (Fig. 4k). The remaining three $On_{tr}$-clusters had similar properties to $C_{15,18}$ (Fig. 5c, f, $C_{14,16,17}$). By contrast, the four $On_{sus}$-clusters ($C_{19-22}$, (Fig. 5g–i) has larger responses to white compared to coloured light (e.g. $C_{20}$ in Fig. 5g, cf. Supplemental Fig. S3 for $C_{19,21,22}$), and their spectral tuning functions were broader than a single opsin in isolation.

To explore how On- and Off-responses encode wavelength information we computed two indices (Methods): The spectral dominance index compares a cluster's largest colour step response to its 100% white-step response, such that −1 and 1 indicate exclusive response to white and colour steps, respectively, while 0 indicates equal responses. Conversely, the spectral tuning index compares the width of a cluster's spectral tuning function to that of its spectrally nearest log-opsin template, such that −1 and 1 respectively indicate infinitely narrow or broad tuning functions, while 0 indicates that the cluster's tuning matches that of a single opsin. Together these two indices revealed systematic differences in how On- and Off-responses encode spectral information. For On-responses, spectral dominance and spectral tuning were strongly correlated, and spanned much of the possible coding space (Fig. 5j, cf. Supplemental Fig. S6a). In the lower left quadrant, the four $On_{sus}$ clusters exhibited spectrally broad and white-dominant responses, while in the upper right quadrant, several of the $On_{tr}$ clusters had some of the most spectral-dominant and/or narrow responses. The remaining Off and OnOff clusters lie between

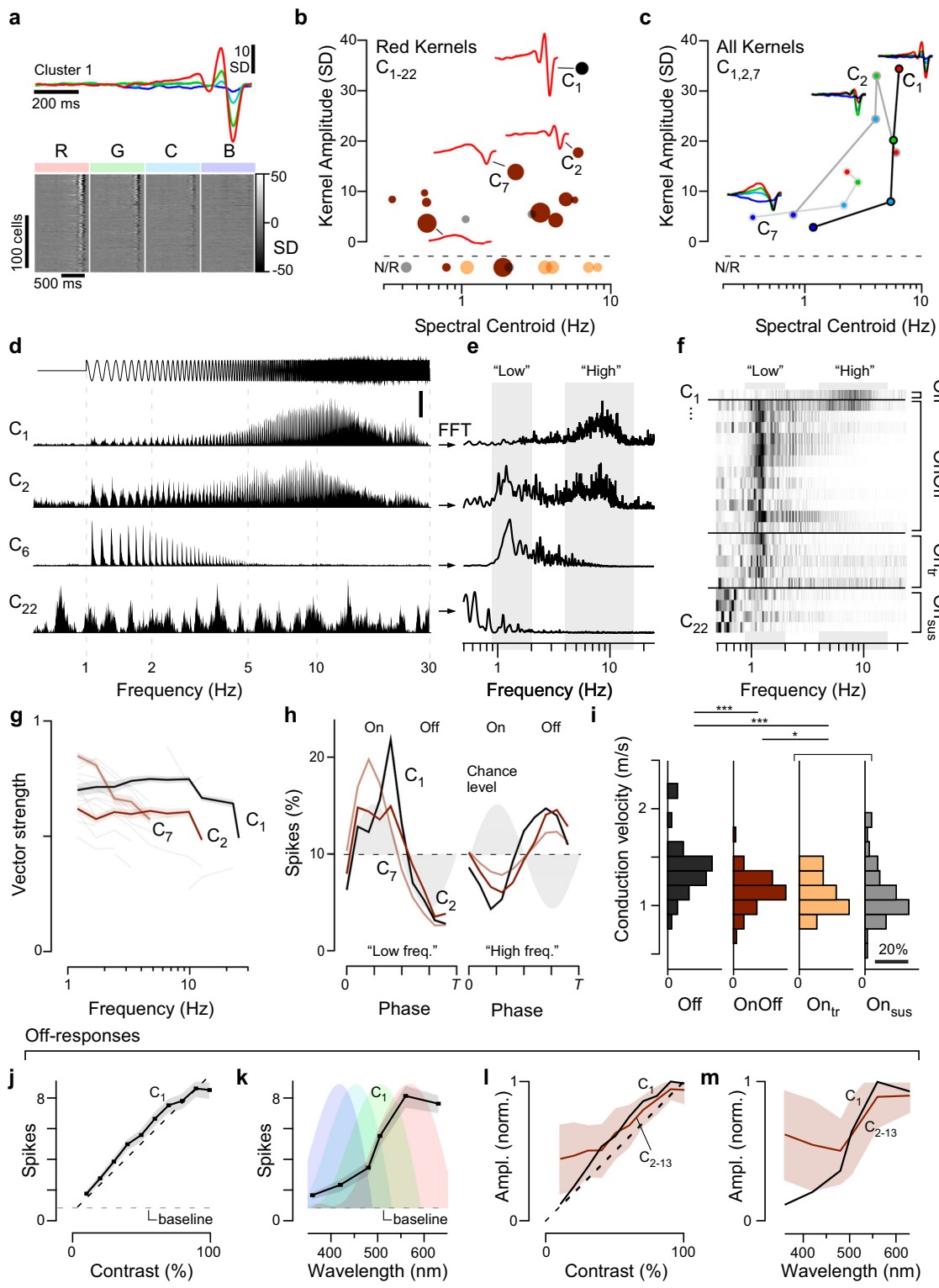

these extremes. By contrast, Off-responses generally fell around the origin of both indices, indicating no preference for white or coloured light, and spectral tuning consistent with drive from a single opsin (Fig. 5k, cf. Supplemental Fig. S6b). Accordingly, On-circuits had spectrally distinct responses, while Off-circuits exhibited low spectral diversity. This difference was also illustrated by spectral tuning functions for On-responses (Fig. 5l) compared to the tunings of Off-responses (Fig. 4m). Moreover, most clusters had markedly sub-linear On- contrast-response functions to white light (Fig. 5m), rather than the more linear contrast-response functions of Off-circuits (cf. Fig. 4l).

The On$_{sus}$ group was an exception to this rule, with supra-linear contrast-response functions.

Taken together, it therefore appears that the chick retina disproportionately leverages Off-circuits to encode rapid achromatic contrast (Fig. 4), and On-circuits to encode wavelength (Fig. 5).

### OnOff channels multiplex spectral and temporal information

Building on the foregoing observations, we asked what happens when the chick's functionally distinct On- and Off-circuits are combined to form OnOff-channels. The combination might allow emergence of new

**Fig. 4 | Off-responses encode achromatic temporal contrast. a** Spectral kernels of the most Off-dominated cluster $C_1$, showing the cluster mean (top) and a heat-map of all constituent cells' kernels (bottom), as indicated. **b, c** comparison of kernel amplitudes and kinetics (spectral centroids, Methods), here shown for 'red' kernels of all clusters (**b**), and for the full complement of R, G, C, B kernels for three exemplary clusters (**c**). **d–f** Mean chirp-responses of four example clusters as indicated (**d**), their corresponding area normalised magnitude squared Fourier transform (**e**) and heatmap of all clusters' Fourier transforms (**f**). "Low" and "High" frequency windows that were used as the basis of computing a High Frequency Index (Methods, Supplemental Fig. 5a, c, d) and for (**i**) are shaded into the background. **g** Degree of phase locking at different frequencies for all clusters, quantified as vector strength (means ± s.e.m.). Three exemplary clusters are highlighted with solid lines, the remainder of clusters is plotted faintly in the background. Non-significant entries (Methods) are not shown. **h** Area-normalised phase histograms (Methods) of three example clusters across two frequency windows as indicated in

(**e, f**) (for all clusters, see Supplemental Fig. 5b). Background shading indicates the phase of the light. **i** Histograms of axonal conduction velocities computed from electrical images for the four cluster-groups as indicated (cf. Fig. 1l). Wilcoxon Rank Sum Test, 1 tailed: Off vs OnOff: $p < 0.001$; Off vs. On$_{(tr+sus)}$: $p < 0.001$; OnOff vs. On$_{(tr+sus)}$: $p = 0.041$; (For the purposes of statistical comparison, On$_{tr}$ and On$_{sus}$ data were combined in view of the relatively small number of On$_{tr}$ cells that yielded a reliable electrical image due to their generally very low spike counts). **j, k** Cluster $C_1$ mean ± SEM number of spikes elicited when probed with the WS (**j**) and CS (**k**) stimuli. The background shading in (**k**) indicate the log-transformed spectral sensitivity functions of the chicks' four cone-opsins—note match of the $C_1$ tuning to the 'red' (LWS) opsin. **l, m** as (**j, k**), respectively, for all clusters that showed a strong off-component. Shown are means ± SD across all peak-normalised cluster means (Methods) allocated to the OnOff group $C_{2–13}$ (brown), and the corresponding normalised entries for the only cluster in the Off group $C_1$ (black). Source data are provided as a Source Data file.

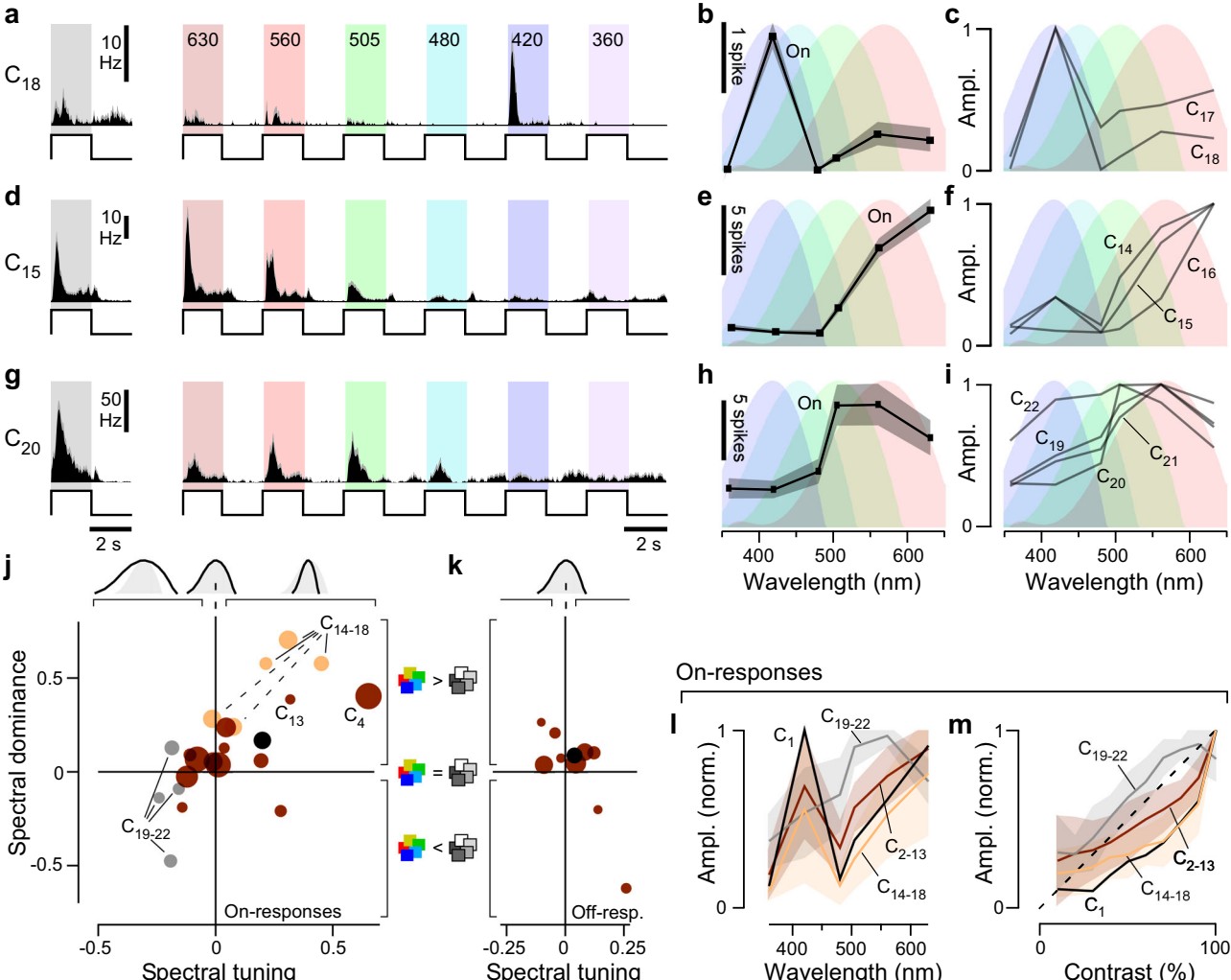

**Fig. 5 | On-responses encode wavelength information. a–i** Mean ± SEM spike-responses to the 100% WS and all six CS stimuli of On-clusters as indicated, showing spike-histograms (**a, d, g**), spike counts per On-response during CS stimulation (**b, e, h**), and normalised mean response amplitudes of all On-clusters that exhibited blue- (**c**) or red-dominated tunings (**f**), and for those that are broadly tuned (**i**). **j, k** Relationship of 'spectral dominance' and 'spectral tuning' (Methods) for all clusters' On- (**j**) and, where applicable, Off-responses (**k**). cf. Supplemental Fig. 6a, b

for a cell-wise plot of the same data. Symbols above the plots indicate spectral width (from left: broader-, equal-, narrower-than-opsin), while symbols to the side indicate a dominant response to 'coloured' (top) or 'white' (bottom) stimulation. **l, m** As Fig. 4m, l, respectively, but here shown for On-responses (means ± SD). Note that unlike for Off-responses (Fig. 4), the spectral tuning of On-responses could generally not be captured by a single opsin (**l**), and contrast-response functions were generally non-linear (**m**). Source data are provided as a Source Data file.

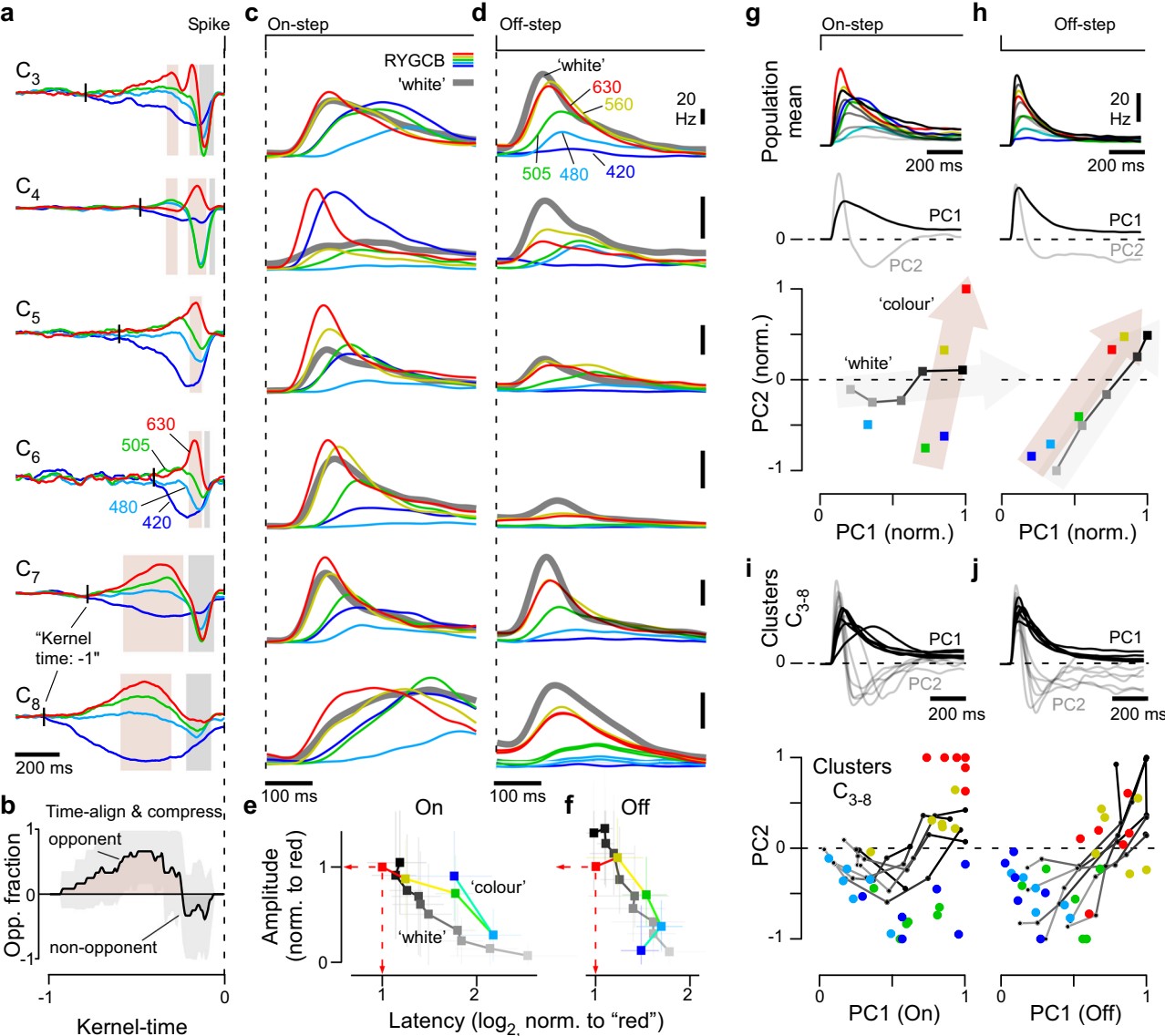

**Fig. 6 | OnOff circuits multiplex spectral and temporal information. a** Example spectral kernels for six of the twelve OnOff clusters as indicated. For each cluster, shadings indicate the parts of the kernels that were classed as either colour opponent (brown) or non-opponent (grey, Methods). The timepoint of each cluster's onset is indicated by a short vertical line (kernel time: −1). **b** Mean ± SD 'kernel-opponency over normalised time' for all twelve OnOff clusters $C_{2-13}$, based on the temporal and opponency measures indicated in (**a**); see Methods for details. On average, OnOff cluster kernels tended to be colour opponent over long timescales (1, brown), but converged onto a non-opponent (−1, brown) period on short timescales (i.e. closer in time to the spike event at kernel time 0). The time-normalisation was required in view of the very different overall kinetic regimes across clusters. **c**–**f** Mean On- (**c**, **e**) and Off- (**d**, **f**) step responses as indicated for the same OnOff clusters listed in (**a**), illustrating some of their diversity in kinetics and amplitudes. Note that for On-, but not Off-, red-responses tended to exhibit the largest amplitudes and shortest latencies. The summary plots in (**e**) and (**f**) show the mean ± SD response amplitudes and latencies of all CS and WS responses for all twelve OnOff clusters, each normalised to their respective red response. **g**, **h** Principal component analysis (PCA) of average On- (**g**) and Off- (**h**) WS and CS responses across all twelve OnOff clusters. Top: Traces going into the PCA, which comprised all CS responses except UV, which was generally weak, and 5 of the 10 WS responses (every second contrast value, i.e., 100, 80, 60% etc.). middle: First and second principal component, as indicated, and bottom: Peak-normalised loadings (Methods) of each step response onto the two components. Note that for On-, but not Off-, WS and CS step responses followed approximately orthogonal trajectories in this PC space (indicated by the shaded arrows). **i**, **j** (as **g**, **h**), but computed separately for the six OnOff clusters shown across (**a**, **c**, **d**). By and large, individual OnOff clusters behaved similarly to the population means (**g**, **h**). Source data are provided as a Source Data file.

functions but degrade one or both incoming messages. Alternatively, if On- and Off-signals remain partly decodable—for example in time—the new channel could encode aspects of both sets of incoming information. Such multiplexed signal transmission might allow efficient use of the optic nerve's limited bandwidth, as in communications technology[51].

Accordingly, we explored the possibility that OnOff-channels in the chick retina might be simultaneously informative about both spectral and temporal contrast (i.e. "colour" and "greyscale").

Evidence for multiplexing comes from the spectral kernels of OnOff cells (Fig. 6a). The full kernels were colour opponent in all OnOff clusters ($C_{2-13}$), but their opponency was usually time-dependent with the spectral kernels converging to a non-opponent Off-signal in the final ~100 ms preceding the spike. This effect is illustrated by cluster $C_7$ (Fig. 6a, fifth entry): at ~500 ms preceding the spike, the cluster is 'red/green'-On 'blue'-Off opponent (shaded in brown), but as the blue-kernel was monophasic (i.e. 'slow') and the red/green-kernels were biphasic (i.e. 'fast'), all three converged to yield an achromatic late Off-period

immediately preceding the spike (shaded in grey). Thus, cluster $C_7$ combines "colour" and "greyscale"-information, segregated in time.

To systematically quantify this property, we defined colour-opponent (+1) and non-opponent (−1) phases in each OnOff cluster's spectral kernels (Fig. 6a, brown and grey shadings, respectively). We also normalised each cluster's kernels in time (with "−1" and "0" indicating the timepoints where the kernels' first exceed a minimum threshold amplitude, and the time of the spike, respectively). For all twelve OnOff clusters we then computed the mean ± SD 'opponency fraction' over normalised time (Fig. 6b; 1 and −1 on the y-axis denoting that all clusters were opponent or non-opponent, respectively, Methods). This confirmed the systematic nature of the spectro-temporal responses: On average, spectral kernels were colour opponent over long time scales, but non-opponent over short time scales.

Even though OnOff clusters differed by nearly an order of magnitude in their overall kernel kinetics (Fig. 6a, cf. Supplemental Fig. S3, Fig. 5b, c), within each cluster, kinetic order was stereotyped: Red-kernels tended to be the fastest, followed by green-, then cyan-, and finally blue. Accordingly, most OnOff clusters encoded a similar spectral hierarchy over different time-scales. Clusters $C_{2-4}$ were fast at all four wavelengths, $C_{5,6}$ intermediate and $C_{7-12}$ were remarkably slow. The achromatic and strongly Off-dominated cluster $C_1$ fitted the fast extreme of this pattern, as the fast non-opponent fraction of the kernel was retained but the slow opponent fraction lost (Supplemental Fig. S3). Overall, the analysis of spectral kernels suggests that OnOff channels simultaneously encode slow spectral and fast achromatic information, with the two sets of features being segregated, and thus decodable, by their relative timings.

### "Time-dependent opponency" in OnOff channels emerges from differential spectral integration across On- and Off-circuits

We next asked how these time-wavelength features might relate to the properties of the On- and Off-circuits in isolation. To this end, we analysed the OnOff clusters' On- and Off-responses to 'white' and 'coloured' steps of light (Fig. 6c–f). In line with corresponding observations at the level of the kernels, this revealed a systematic wavelength-dependence of clusters' step responses: For both On- and Off-transitions, the largest amplitudes tended to occur for red, followed in turn by yellow, green, and cyan. For the Off-channel, this trend extended to blue, while for On, blue-responses tended to be larger than cyan responses. Together, these spectral responses accounted for the previously observed biphasic and monophasic spectral tuning functions of the On- and Off-channel, respectively (On: Fig. 5l, Off: Fig. 4m).

Step responses differed in their latencies as well as their amplitudes (Fig. 6c, d). For example, red responses tended to be large, with short latencies, while cyan responses were smaller and delayed. An inverse link between response amplitude and latency is expected, but the variance across these two parameters was sometimes substantial. For example, the latencies of red- and green-On responses of cluster $C_6$ differed by more than 60 milliseconds (red: 104 ms, green: 167 ms) despite their almost identical slopes (Fig. 6c, fourth entry). By comparison, the On-response to the 100% white step was intermediate at 110 ms. In fact, within the On-channel, the latencies of red-responses tended to be slightly below those of the corresponding 100% contrast white step responses. Conversely, in the Off channel the white response was generally dominant (Fig. 6d).

To explore if and how these types of amplitude and kinetic differences might encode stimulus wavelength and/or intensity, we quantified the OnOff clusters' 'colour' and 'white' step-response amplitudes and latencies and normalised each relative to their respective red-response. (Fig. 6e, f). This revealed that for the On-, but not for the Off-channel, the combination of these two simple metrics alone sufficed to substantially disentangle wavelength from intensity information. Red-responses were systematically faster than white-

responses, allowing their detection by latency alone (Fig. 6e), whereas, green- and blue-, but not cyan- or yellow-, could be distinguished from the 'white' contrast series by their long latencies, despite their large amplitudes. While white-responses increased in latency while dropping in amplitude with decreasing contrast, blue- and green-On responses had almost equal amplitudes to red-responses but a nearly two-fold greater latency. In contrast, the Off-responses to 'coloured' and 'white'-steps had similar amplitudes and latencies, precluding their differentiation by these metrics (Fig. 6f).

Beyond amplitude and latency, step responses of the OnOff cells differed in their overall temporal envelopes (i.e. the detailed time-courses of step-responses). Accordingly, to capture amplitude and kinetic differences more comprehensively across 'coloured' and 'white'-step responses we used Principal Component Analysis (PCA). We first averaged step responses to all red-, yellow-, green-, cyan- and blue- steps as well as to the 100, 80, 60, 40 and 20% white contrast steps (Fig. 6g, h, top), which as expected reproduced the amplitude and kinetic features discussed previously. From these averages, we performed PCA separately on On- and Off-responses, both of which yielded a relatively slow first component followed by a faster second component that together captured >99% of the total variance (Fig. 6g, h, middle). We then projected each response's loading onto these two components (Fig. 6g, h, bottom). As predicted from our analysis of amplitudes and latencies alone (Fig. 6e, f), this highlighted systematic differences in the encoding of 'coloured' versus 'white' stimuli for the On-channel, but a common encoding scheme for the Off-channel. For On-responses, the white contrast series was almost completely captured by the first principal component, while colour steps exhibited an approximately fixed loading onto PC1 but wavelength-dependent differences onto PC2. As expected, for Off-responses colour and white steps followed essentially the same trajectory in PC-space.

Finally, to establish the generality of this encoding strategy, we performed PCA separately for a subset of individual OnOff-clusters ($C_{3-8}$) (Fig. 6i, j). In each case, ensuring a common polarity of the first two components (Fig. 6i, j, top), we normalised the loadings onto the two PCs between −1 and 1 to enable side-by-side comparison (Fig. 6i, j, bottom). By and large, this strategy recapitulated the same distribution of loadings in PC space across coloured and white steps. White-On responses mostly followed PC1 with generally only weak loadings onto PC2, while conversely, red-, yellow, green- and blue-On-responses required a relatively fixed loading onto PC1 but systematically distinct loadings onto PC2. In contrast, as before, coloured, and white Off-responses followed a common, intermixed trajectory.

Taken together, we conclude that in the chick retina, On- and Off-channels predominately encode slow-spectral and fast-achromatic information, respectively, while a majority of OnOff-channels simultaneously capture both types of information.

## Discussion

We have shown that the spiking output from the chick retina comprises multiple kinetically and spectrally diverse OnOff-channels, alongside sparser populations of fast achromatic Off-, and slow chromatic On-channels. These correlations between polarity, kinetics, and wavelength selectivity (Fig. 3a–c) imply that there is a general organising principle in this avian retina, where Off- and On-channels represent fundamentally different aspects of the visual scene namely: time and colour.

### Pathway splitting in non-mammalian retinas

The division of the visual signal into On- and Off-pathways at the retina's first synapse is a fundamental and ancient[52] feature of vertebrate vision[4]. The division reduces energy requirements and increases the dynamic range[53] so it is not surprising that sensory systems should balance On- and Off-channels, as exemplified in mammal RGCs[2] (cf. Fig. 3d–f). Well-segregated and coordinated On- and Off-channels

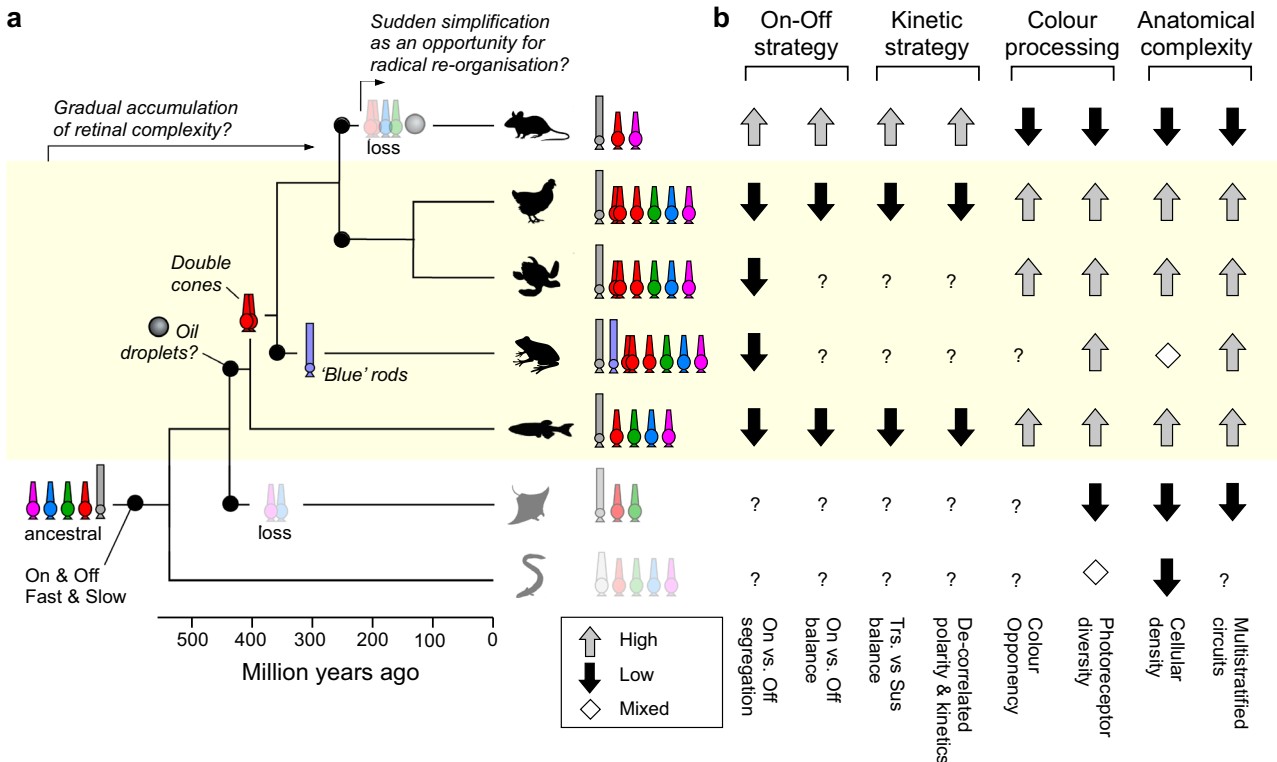

**Fig. 7 | Retinal organisation in the context of vertebrate phylogeny.**
**a** Approximate vertebrate phylogeny, indicating key events in the evolution of
retinal circuits (based on refs. 29, 30): The split of the visual signal into On- and Off-
circuits likely predates the divergence of jawed from jawless species in the early
Cambrian[93], and probably around the same time, an ancestral RH-opsin expressing
photoreceptor gave rise to vertebrate rods and green-cones[94] (but see ref. 30). Oil
droplets are absent in mammals and elasmobranchs, but present in species of fish,
amphibians, reptiles and birds, and non-placental mammals[42]—accordingly the
mostly likely origin of oil droplets was between the split of cartilaginous and teleost
fish, later followed by their loss in early placental mammals. Amphibians further
evolved 'blue' rods, while early tetrapods evolved the double cone that is still

present in extant birds. By contrast, eutherian mammals lost ancestral SWS2 and
RH2 single cones, double cones, and oil droplets, while cartilaginous fish lost SWS1
and SWS2 cones (and sharks also lost LWS cones), and their ancestors never had
double cones or oil droplets. The resultant maximal cone-complement is indicated
for each lineage: RH1 rods (grey), LWS/double cones (red), RH2 cones (green),
SWS2 cones (blue), SWS1 cones (pink) and SWS2 rods (faded blue). **b** Approximate
typical anatomical and functional arrangement of retinal circuits in each lineage.
The question marks indicate entries that have not been studied. References for
each entry are provided in the relevant sections throughout the main text. Frog
silhouette in (**a**) from silhouettegarden.com.

also emerge de novo in retina-inspired computational models aiming
to capture as much information as possible from natural scenes[54,55].
Why, then, do chicks not comply with this arrangement?

One partial explanation might be that beyond efficient use of
neural bandwidth and energy, other aspects of visual processing
benefit from combining On- and Off-signals. Specifically, encoding
wavelength independent of intensity, which is an essential feature of
colour vison, requires opponency between spectrally distinct On-
and Off-inputs[29]. One example of an OnOff chromatic opponent
channel is the primate small bistratified RGC[56,57], which has two
notable parallels to the multiple colour opponent OnOff cells of
chicks. First, as in chicks, the Off-circuit is driven by LWS cones, and
second, the blue On- and yellow Off-circuits have different time-
courses: On is slow, but Off is fast[56]. In primates, this feature is
probably inherited from corresponding kinetic differences between
SWS1 and LWS cones[58]. It is unknown if also in chicks the different
spectral photoreceptors have systematically different kinetics—but
if they do, it might partly account for the correspondingly sys-
tematic kinetic differences in spectral kernels observed at the level
of RGCs (Fig. 6a).

Nevertheless, with ~67% of all recorded cells, the numerical
abundance of chick colour-opponent OnOff RGCs is superficially at
odds with efficient coding theory[59–61]. Spectral variance makes up only
a small fraction of information in natural scenes, which are dominated
by achromatic contrasts[5]. Correspondingly[59–61], visual systems

including our own are thought to prioritise the encoding of intensity
rather than wavelength[8,62]. This implies that beyond wavelength, OnOff
RGCs in chicks might encode other aspects of the visual scene, such as
time: while the spectral and kinetic order of OnOff clusters' kernels was
remarkably consistent (from 'red and fast', to 'blue and slow'), the
overall response kinetics of these clusters varied by more than a log
unit in speed. The different OnOff clusters therefore systematically
encoded similar sets of spectral and kinetic contrasts, but for a range
of temporal regimes.

**Why are chick and mammal retinas so different?.** The systematic
functional differences in the retinal organisation of chicks, and per-
haps of other vertebrate lineages, from that of mammals (Fig. 3)
implies that different clades have evolved divergent strategies for
communicating visual information from the eye to the brain (Fig. 7).
Exploring these differences, their computational consequences, and
how they may have come about, will be important in the future, but for
now it may be useful to posit one possible avenue of exploration;
namely that birds, reptiles, amphibians, and fish differ from mammals
in their complements of photoreceptors (Fig. 7a). While some lineages
in each of the former clades retain the full complement of four
ancestral single cones[63]—often elaborating them in various ways—early
mammals lost their SWS2 and RH2 cones[29,30], so that typical mamma-
lian retinas are driven by two cone inputs, but many non-mammalian
retinas are driven by four or more.

This systematic difference in inputs may lead to different retinal processing challenges. Perhaps the complex interplay of time and wavelength coding in the chick is a consequence of their relatively complex input system that is carried over from ancestral vertebrates, while the loss of two cones more than 200 million years ago, alongside −presumably−freeing up a substantial diversity of now disused inner retinal circuits, left mammals with an opportunity to evolve new, powerful processing strategies that were previously precluded.

In support, mammalian retinas also differ from those of birds, reptiles, amphibians, and fish in other ways (Fig. 7b). For example, their neuronal density is substantially lower[2,34] (with the exception of salamanders[19], but not frogs[64]). The only other major vertebrate lineage with consistently low cellular density retinas are elasmobranchs[65] (sharks, rays, and skates), which, like mammals, have lost two or more of the four ancestral cones[29]. Mammals[49,66] and elasmobranchs[67] are also unusual among vertebrates in that most of their bipolar cells and RGCs are mono- or at most bistratified, rather than routinely multi-stratified as is the case in birds[20,32], reptiles[68], amphibians[69], and fish[10,70,71]. In view of the well-documented links between inner retinal depth, response polarity, and kinetics[3,5,6,12,36,72], these anatomical differences may directly feed into the observed functional differences. Future functional exploration of other vertebrate retinas, especially those that like mammals have lost a subset of their original photoreceptor complement, may be instructive.

### The avian double cone as the input to the fast, achromatic Off-circuit?

Bird and reptile eyes are unique having 'double cones' that are distinct from the full complement of ancestral single cones[29,32,46]. Double cones are made up of two tightly associated cells: a principal and an accessory member, which are independently wired into outer retinal circuits[32]. Both members express the same 'red' LWS opsin that is also found in the ancestral LWS single cones. However, unlike LWS single cones of other species, which are generally associated with achromatic processing[30,31,40], direct recordings from either member of avian double cones have not been achieved, leaving insights as to their functions speculative[73]. In general, their numerical abundance in the periphery[74], but absence from the fovea[75], hints at a key role in finely resolved temporal rather than spatial processing[10,76,77].

In chick, the only robust responses to fast achromatic flicker occurred in the Off channel, whose spectral tuning was consistent with near-exclusive drive from an LWS-expressing photoreceptor system (Fig. 4). The simplest explanation for these observations is that the chick's rapid and achromatic Off-circuits are driven by either or both members of the double cone, and/or the red-single cone. In tentative agreement, at least two anatomical types of chick bipolar cells receive exclusive direct input from the principal member of double cones alongside inputs from rods[32]. Both these cells stratify in the upper to middle fraction of the inner retina which is generally associated with Off-dominated processing[5]. Our data therefore lends further credence to the idea that the avian double-cone system might support fast, achromatic vision[46], and add the perhaps surprising notion that this signal appears to be exclusively carried by Off-circuits.

### On for 'colour', Off as a common reference

To distinguish wavelength from intensity, circuits for colour vision use colour opponency as their fundamental 'currency'[29]. However, beyond a basic requirement of combining spectrally distinct On-and Off-signals, there are many options to build an opponent circuit. A short-wavelength On-pathway could be combined with a long-wavelength Off-pathway, but it is equally plausible to do the reverse, i.e., to combine short-Off with long-On. And yet, overwhelmingly, vertebrate circuits for colour vision, including in mammals, appear to favour the former[9,11,29,30,39,78]. Second, when more than one colour opponent axis is established, the second axis could either oppose two entirely new

wavelength ranges, or it could reuse one of the two signals from the first axis. Here, the spectral heterogeneity of the On-channels (Fig. 5), but homogeneity of the Off-channels (Fig. 4), suggests that chicks do the latter: They systematically oppose their spectrally diverse On-signals to a common, LWS-driven Off reference. Similarly, the zebrafish brain appears to be dominated by spectrally narrow and diverse On-signals, but spectrally homogenous, broad and LWS-shaped Off-signals[13].

### Cortex-like hue-coding in the avian retina?.

Despite lacking overt signs of colour opponency, many cells were 'spectrally selective' in that they exhibited sharper-than-opsin spectral tuning and a preference for spectrally narrow over 'white' light (Fig. 5). Sharper-than-opsin tuning can, in principle, be achieved by rectification of a spectrally broader, non-opponent drive. However, in that case the cell would nevertheless be expected to respond strongly to 'white' stimulation, which was not the case for many of the spectrally narrow clusters (Fig. 5j). Alternatively, narrow tuning could be built by rectifying an already opponent input. For example, a hypothetical RGC rectifying an incoming drive from a colour opponent BC[12,36,79] such that only the On-lobe of the opponency persists could readily account for the profusion of 'colour-selective' On-RGCs in our dataset. Neurons with similar properties have long been discussed as part of the colour vision machinery of the primate cortex, where they are usually referred to as hue-selective[80]. Similarly, narrower-than-opsin On-responses also exist in the zebrafish tectum[13,15] and in the mushroom bodies of butterflies[81]. To our knowledge, they have not been described in a vertebrate retina.

## Experimental model and subject details

### Animals

All procedures were performed in accordance with the UK Animals (Scientific Procedures) act 1986 and approved by the animal welfare committee of the University of Sussex. Male chicks, breed Shaver Brown, aged between 1 and 14-days post hatching were obtained from Joice and Hill (part of Hendrix Genetics, Peterborough, UK) and kept in a specifically designed cage in the university's animal facility. Food (chick crumb) was provided ad libitum and elements for facilitating play behaviour were provided. Chicks were kept on a 12 h:12 h light:-dark cycle, and never kept in isolation for prolonged periods of time. All experiments were performed on 1–14 day old chicks.

### Ringer solution

Ringer solution for the experiment was based on ref. 82, calibrated to 0.331 osmol. In moles per l: 0.1 NaCl, 0.006 KCl, 0.002 $MgSO_4$, 0.001 $CaCl_2$, 0.03 $NHCO_3$, 0.001 $NaH_2PO_4$, 0.05 sucrose. Two litres of ringer solution were freshly prepared for each experiment. The solution was bubbled with carboxygen (95% $O_2$ / 5% $CO_2$) for at least one hour before experiments and heated to 37 °C. A second batch of the same ringer solution, however with added 0.4 mM $MgSO_4$, was prepared for dissection.

### Enucleation and dissection

Chicks were dark adapted for at least 12 h overnight. In total darkness using infrared goggles, chicks were sacrificed by cervical dislocation followed cutting of the aorta. Eyes were enucleated by first cutting the eyelid around the cornea, followed by a single anterior leading cut between eyes and beak. Using curved forceps (FST 11652-10, FST Heidelberg, Germany), the eyes were then lifted, and the optic nerve was cut using scissors with a partially blunt tip (FST 14083-08). In the following the remaining muscle tissue around the eyes were removed and lifted from the skull. The eyes were cut two times proximal to the edge of the cornea using pointed scissors (FST 15017-10) and transferred into two bottles containing preheated, high-magnesium oxygenated ringer solution at 37 °C. The bottles were light sealed. Eyes were

transferred to the experimental site for retinal dissection. Total time for the above was ~15 min.

Dissection followed procedures detailed in ref. 82. However, all steps were performed under infrared light using night vision googles (PSV-14, ACT in Black, Luxembourg), and in a high-magnesium dissection ringer solution. The following steps were performed in a petri dish unless otherwise stated:

1. Removal of the cornea from the eyeball with as few cuts around the horizon as possible (FST 15017-10).
2. Cutting the eyeball along the dorsal – ventral axis using two cuts from opposite sides.
3. Removing the vitreous from the eyeball using forceps.
4. Cutting a ~4×4 mm piece out of the central area of the dorsal hemisphere.
5. Transfer of this piece onto a filter paper, with the RGCs facing the filter paper and the remaining sclera facing up.
6. The filter paper and tissue were then transferred onto a kitchen roll paper to draw solution, which flattened the tissue and aided attachment to the filter paper.
7. The remaining sclera, choroid and retinal pigment epithelium were removed from the retina by using forceps to peel those layers off the retina, which remained attached to the filter paper.
8. The filter paper and retina were transferred back to the petri dish.
9. The retina was removed from the filter paper using forceps.
10. A smaller, about 2.5 mm² piece was cut. The retina would commonly get folded at a few places during step 6 due to flatting of what normally is a curved tissue. In this step, an area without folds was chosen. Folded parts of the retina were avoided even if this resulted in smaller preparation. In addition, corners in the tissue were rounded off since these sites tended to trigger the degenerative waves.
11. The retina was transferred to the MEA chamber using a spoon. The tissue was guided onto the spoon using forceps and was continuously guided while being on the spoon to avoid strong movement of the tissue during transfer.
12. In the MEA chamber the corners of the tissue were cut. This was done, to remove parts of the tissue that had been damaged by the forceps during tissue guiding steps.
13. The tissue was placed onto the electrode array with a fine paintbrush.
14. The MEA chamber was dried using kitchen roll to suck out the ringer solution. As soon as the tissue was exposed to the air new ringer solution was added directly on top of the tissue. This would normally attach the tissue to the MEA.
15. The MEA chamber was connected to the head stage.
16. The tissue was left to further attach to the MEA for ~30 min.
17. The tissue was perfused throughout the experiment using the low magnesium-ringer solution. The MEA chamber was heated to 37 °C and the perfusion solution was preheated inline to the same temperature.

Throughout, we focussed on the dorsal (cf. Supplemental Fig. S1e) retina due to its approximately representative RGCs density for the whole retina[20], and because it remains far from the pecten, area centralis and optic disc where the large number of axon bundles and thick inner limiting membrane made the recording of reliable signals increasingly difficult.

## Method details
### Light stimulation
Since our multielectrode array was opaque, all light stimuli were delivered from the photoreceptor side by a combination of two custom-modified 3-channel light projection engines ("Lightcrafters") that together were driven with six spectral LEDs (UV: 360 nm, LED365-06Z, 20 nW; Blue, B: 420 nm, ROHS 247–1757, 65 nW; Cyan, C: 480 nm,

ROHS 810–0492, 60 nW; Green, G: 505 nm, ROHS 769–3551, 65 nW; Yellow, Y: 560 nm, RoHS 247–1735, 59 nW; Red, R: 630 nm, RoHS 904–7367, 100 nW) as described previously[83]. This approximately corresponds to low photopic conditions. All stimuli used in this study were applied as widefield, exceeding the active area of the multi-electrode array (2.67 mm²). The framerates of the stimulators were 60 Hz. LEDs 420 nm–630 nm were bandpass filtered using Edmund Optics' 65–137, 65–145, 34–506, 88-011, 65–166 and combined using Chroma Optics beam splitters: NC474265, AT455DC, T550lpxr and T600lpxr. Four types of full-field stimuli were used: (i) flashes of light with all 6 LEDs driven together, at ten different contrast levels ('white steps' WS, 100-10%, 2 s On, 2 s Off – here "100% white" meant that all six LEDs were maximally active at the same time), (ii) flashes of light at each of the six wavelengths individually ('colour steps', CS, 2 s On, 2 s Off), (ii) a 100% contrast white' chirp stimulus[68] (exponentially accelerating from 1–30 Hz over 30 s), and (iv) a binary dense and spectrally flat white noise, in which four of the six LEDs (R, G, C, B) were flickered independently in a known binary M-sequence at 20 Hz for 20 min. This allowed us to estimate spike-triggered averages (STA) per LED and cell, hereafter referred to as spectral kernels (SK, below).

### Multielectrode array recordings
MEA recordings were performed on a BIOCAM X platform produced by 3Brain AG, Wädenswil, Switzerland. The chip was Arena HD-MEA. It consists of 4,096 electrodes in a 2.67 × 2.67 mm square area (64*64 21 µm electrodes with 42 µm interelectrode spacing). The chamber above the array is 7 mm deep and 25 mm in diameter. The software for data acquisition was Brainwave 4. It was run on a bespoke PC provided as part of the MEA by 3Brain. The integration time was set to medium, resulting in a sampling frequency of ~18 kHz.

### Data analysis
Data analysis was performed using custom written Python (v. 3.9.5, Python Software Foundation, Wilmington, Delaware, USA) scripts executed in JuypterLab, IGOR Pro 6.3 (Wavemetrics), Fiji (NIH) and Matlab R2019b / R2020b (Mathworks).

**Spike sorting.** Evaluating the quality of different spike sorters or a specific sorting is an ongoing field of research[84,85]. In this study we faced the specific challenge that most currently available spike sorting algorithms were developed and tested on mammalian or amphibian tissues. However, the cellular density of the chicken retina far exceeds that of the neural tissues commonly used with these types of recordings. Consequently, each electrode likely picked up the extracellular signals from a relatively larger number of neurons, which posed new challenges in spike sorting. In the current absence of ground truth data in recordings from birds or animals with similarly high neuronal densities, we therefore opted for a maximally conservative approach to spike sorting. To this end, as detailed below, we developed and implemented a series of quality measures that rejected the vast majority of initially spike-sorted cells on the array. We also calculated additional sorting quality metrics to rule out systematic biases in our spike sorting results. For example, since we found a high number of OnOff cells, we specifically tested if these units could have been the result of a systematic merging of On and Off cells into one unit. We also tested if other trends in the data were correlated with spike sorting quality. We did not find evidence for this. Nevertheless, further independent studies on bird retinas are necessary to build towards a better understanding of potential biases in spike sorters when sorting data from bird retinas with high cellular densities. We will be pleased to support any such or other projects by sharing our raw recordings without restriction. In view of the substantial size of these datasets (~9 GB per minute of recording time), it is not practical to host these online, and they will therefore be made available upon request to the lead contact.

**Herding Spikes 2**. Herding Spikes 2 (HS2) algorithm[86] was used for spike sorting. HS2 uses a combination of spike location and spike shapes information and performs a mean shift clustering approach to separate spikes coming from different cells. The following parameters were used: clustering_bandwidth: 8; clustering_alpha: 5.5; clustering_n_jobs: −1; clustering_bin_seeding: True; clustering_min_bin_freq: 16; clustering_subset: None; left_cutout_time: 0.3; right_cutout_time: 0.3; detect_threshold: 5; probe_masked_channels: []; probe_inner_radius: 70; probe_neigbor_radius: 90; probe_event_length: 0.26; probe_peak_jitter: 0.2; num_com_centers: 1; maa: 12; ahpthr: 11; out_file_name: HS2_sorted.hdf5; decay_filtering: False; save_all: False; amp_evaluation_time: 0.4; spk_evaluation_time: 1.0; pca_components: 2; pca_whiten: True; freq_min: 300.0; freq_max: 6000.0; filter: True; pre_scale: True; pre_scale_value: 20.0; filter_duplicates: True. After spike sorting, cells that spiked >10 spikes per second on average over a period of one hour were excluded from any further analysis.

### Spike sorting quality metrics

**Circularity Index.** We reasoned that a perfectly spike-sorted cell should have a perfectly circular cloud of spike locations in space. In this case, all principal components across spatial locations should explain the same amount of variance. Correspondingly, we calculated the first two principal components and defined the circularity index as variance explained by PCs1 divided by variance explained by PC2. A circularity index of 1 indicates a perfectly circular cloud of spike locations in space, while higher values indicate spatial skew. The median circularity index of all cells in the dataset was 1.6, with a standard deviation of 1.1. Moreover, the vast majority of cells with circularity indices above 2 were located at the array edge, where the spatial sampling bias will inevitably produce skewed location clouds. Together, the results from this analysis indicate that by and large, spike-locations for single units were well-located in space.

**Waveform similarity.** For each spike-sorted unit we calculated the first principal component across all waveforms and tested the distribution of resultant loadings onto PC1 for unimodality using Hartigan's Dip test[87]. The closer the resultant p value is to 1, the more likely the distribution is unimodal, suggesting that the waveforms originated by a single cell. The median p value for cells in the dataset was 0.88 with a standard deviation of 0.26.

**On vs. Off quality metrics.** To rule out a systematic mix up of On and Off cells into ON-Off cells we divided all spikes driven by colour- and contrast-steps into "On-spikes" and "Off spikes" based on their timing relative to the stimulus. We then split neurons with a waveform similarity below 0.9 (48%) into two "sub-neurons" according to the shapes of their spike waveforms. We used Gaussian mixed modelling clustering (sklearn.mixture.Gaussian, covariance_type = " full") to split the waveforms according to their positions in PC1. Using $X^2$ test for independence (scipy.stats.chi2_contingency) we checked if the number of On or Off responses were independent from the newly created sub-neurons zero and one. This was the case for most cells. We found 324 cells with p values below 1 of which 103 cells with p values smaller than 0.05. Considering that we performed multiple statistical tests, using Bonferroni correction, we found 13 cells with $p$ values smaller than $p = 2.5 \times 10^{-5}$. Importantly, those 13 (out of 3987) "poorly sorted" cells appeared to be randomly distributed across functional clusters, suggesting that they did not drive systematic trends in the data.

The above-listed metrics are illustrated in Supplemental Fig. S2a–c using examples of "good" and "bad" cells.

**Correlation metrics.** To rule out additional possible systematic biases in our dataset we calculated several further quality metrics and checked for other possible correlations that could explain a high number of OnOff cells (Supplemental Fig. 2d–f). For example, we allocated all cells to 10 bins based on their OnOff index and them tested if these bins were systematically related to variations in the circularity index, waveform similarity index or mean amplitude of the spike signal. Since the OnOff-index was not normally distributed we used Spearman's rank (scipy.stats.spearmanr) correlation. This revealed significant (at $p < 0.001$) but very weak correlations with the circularity index (correlation coefficient $\rho = 0.10$), the waveform similarity index ($\rho = -0.07$) and spike amplitudes ($\rho = -0.057$). We also calculated 2D histograms (similar to the one described above for the OnOff index) to survey for other possible correlations between spike sorting metrics and indices presented in this study (OnOff index, best chirp frequency, TrSus-index, kernel opponency index). Other than the abovementioned weak effects, this revealed no clear additional correlations across these metrics.

### Electrical footprints

Spike triggered averages (STA) were computed by reverse correlating spikes from each cell with the voltage signal of all MEA channels within a window of 10 frames before and 40 frames after a spike resulting in 4096 STAs (one for each channel), consisting of 50 frames. For this, all spikes that occurred within the first half hour of a recording were considered. Electrical footprints were computed for a subset of 5 recordings containing 1,978 cells (47% of cells in the dataset).

### Processing electrical images (EIs)

To identify which channels had recorded an electrical signal from a cell, channels were filtered by calculating a signal strength matrix:

$$Esignal = \{max(STA0), max(STA1), \ldots max(STA4096)\}$$
$$\times \{min(STA0), min(STA1), \ldots min(STA4096)\} \quad (1)$$
$$\times \{diff(STA0), diff(STA1), \ldots diff(STA4096)\}$$

The resulting *Esignal* matrix was median-normalised and all channels with a score below 0 were filtered out. The matrix was reshaped according to the MEA array ($64 \times 64$) and channels with a zscore above 0 but without neighbouring channels were filtered out. If the remaining number of channels was above 400 the EI was not further considered. In most cases only a handful of channels would remain (called filtered channels in the following), of which most would show a typical spike waveform in their STA. We next detected and traced the axon, where present, as follows: The channel with the highest *Esignal* value was defined as the spatial origin of the spike $Smin_O$, and the frame with the lowest number of counts in the corresponding STA was defined as the temporal origin of the spike signal $Fmin_O$. The soma of the cell was identified by identifying all filtered channels within a radius of 6 channels around $S_0$. Those channels were not considered for the axon tracing. To trace the axon, an array was created storing x and y positions of all remaining filtered channels and their respective Fmin values. The array was sorted from low to high Fmin. Based on the x and y positions a random_geometric_graph was created using the networkx Python package (https://networkx.org/documentation/stable/index.html). Using the function shortest_path, the shortest path between the origin of the signal at $S_0/Fmin_O$ and $S_{max}/Fmin_{max}$, was calculated. The distance of the part $D_{short}$ was calculated as the total length of the path multiplied by the width of the array in mm.

The speed of the signal in m / s was calculated as:

$$S_{axon} = \frac{D_{short}}{Fmin_{max} \times Freq} \times \frac{1}{1000mm} \quad (2)$$

with Freq corresponding to the reciprocal of the MEA's sampling frequency.

### Estimating RGC numbers from electrical images

An axon could be clearly detected in 1035 (52%) of the 1978 cells for which electrical images were computed, suggesting that these correspond to axon-bearing retinal ganglion cells (RGCs) rather than axonless displaced amacrine cells (dACs). However, this RGC fraction is almost certainly an underestimate because many axons were likely missed due to limited signal to noise in a subset of electrical images. Here, signal to noise directly depends on the number of spikes that go into the computation of the electrical image, meaning that the reliability of axon detection should increase as a function of available spikes, as shown in Fig. 1k. For example, for 782 of the 1978 cells, we obtained less than 1000 spikes, which clearly is insufficient to reliably disambiguate the presence or absence of an axon in this dataset. At around 5000 spikes and above, the number of cells with a detectable axon plateaued at ~87%, suggesting this number as a lower bound for RGCs in our dataset.

### Clustering and initial computation of STAs

Clustering was performed on a dataset containing the functional responses of 'cells' recovered from the spike sorting 'coloured' (CS) and 'white' steps (WS) of light, chirp (Chirp) stimuli, and spectral kernels (SK) derived from the spectral noise (SN) stimulus. In all experiments, the complete CS and WS stimuli were repeated 5 times, the chirp stimulus was repeated 3 times and the SN stimulus ran for 18,000 stimulus frames (20 min) without repeat (using the R, G, C and B LEDs).

In what follows we describe the procedures followed to cluster the data. We clustered using all four of the above stimuli: CS, WS, Chirp and SK.

To determine the spiking rate over time for each cell in response to each of the CS, WS and Chirp stimuli we mapped all spike times onto the time interval spanned by the first repeat of that stimulus and applied kernel density estimation (KDE) using the Matlab routine ksdensity. We used the default probability density function for the KDE, such that the area under the resulting curve is equal to one, thus normalising the spiking rate across cells and stimuli. The KDE was computed at 1000 equally spaced points and a smoothing bandwidth of 0.05 employed for all three stimuli.

Spectral kernels were computed by collecting the stimulus segments preceding each spike to form a spike-triggered ensemble (STE) and taking the mean of this ensemble. We then subtracted the mean raw stimulus, calculated as the mean of all possible stimulus segments. Stimulus segments were calculated over the one second interval preceding each spike ($-1$ s to 0 s inclusive) at 51 evenly spaced time points (0.02 s intervals). Thus, the segments were calculated at a finer temporal resolution than the stimulus (20 Hz = 0.05 s intervals) resulting in smoother and more detailed SKs (since a spike may occur at any time within a given stimulus window, a higher temporal resolution results in different stimulus segments depending upon the spike's location within that window). For the mean raw stimulus, the mean was taken over all stimulus segments during the interval over which the stimulus was played, segment initiation times ranging from the earliest to the latest possible time that will allow a full stimulus segment, in intervals of 0.02 s. The spectral kernels were further normalised, by subtracting the mean and dividing by the standard deviation of the SK in the $[-2,-1]$ second interval preceding each spike, where this SK is calculated in the same way as the original spectral kernels. This allows us to determine spectral kernel quality (see below) in those cases where the spike count is low. Note that the spectral kernels represent the linear part of each cell's response, meaning that non-linear effects (for example On-Off behaviour) are not captured. For simplicity, we here focussed only on the linear aspects of noise-responses, and instead draw information about nonlinear properties of stimulus encoding from the responses to the complementary step-stimuli (see above).

Cells with low-quality responses to all four stimuli were identified and removed from the data set, cells with a high-quality response to at least one stimulus being retained in all cases. The quality of response to the CS, WS and Chirp stimuli was determined using the signal-to-noise ratio quality index: $\mathrm{QI} = \mathrm{Var}\left[\langle \boldsymbol{C} \rangle_r\right]_t / \langle \mathrm{Var}[\boldsymbol{C}]_t \rangle_r$, where $C$ is the $T$ by $R$ response matrix (time samples by stimulus repetitions), and $\langle \cdot \rangle_x$ and $\mathrm{Var}[\cdot]_x$ denote the mean and variance respectively across the indicated dimension, $x \in \{r, t\}$ (see ref. [38]). This quality index was applied to the KDE-derived spiking rates, where KDE was applied to each repeat separately rather than mapping all spikes onto the first repeat as for the computation of the spiking rates used for clustering (described above). A quality threshold of 0.4 was chosen, below which CS, WS and Chirp responses were judged to be of poor quality. We calculated the standard deviation in the light intensity over time for each stimulus colour in the kernel (R, G, C and B). The kernel quality of each cell was defined as the maximum standard deviation across the four LEDs. A kernel quality threshold of 2.5 was chosen, below which kernels were judged to be of poor quality. The raw data set was of size $n = 60{,}713$, spread across 17 separate experiments. Following quality control, the data set was of size: $n = 4230$ (7.00% (3 s.f.) of the original).

We used principal component analysis (PCA) to reduce the dimensions of the problem prior to clustering. PCA was performed using the Matlab routine **pca** (default settings). We applied PCA separately to each of the 10 segments of the WS stimulus (100%, 90%, …10% contrast segments), to each of the 10 WSs, the Chirp stimulus, each of the 6 CSs, and four SKs. We retained the minimum number of principal components necessary to explain ≥50% of the variance. The resulting 23 'scores' matrices were then concatenated into a single matrix ready for clustering. The following numbers of principal components were used – CS: 24 components in total (3 R components, 3 Y components, 4 G components, 5 C components, 2 B components and 7 U components); WS: 61 components in total ([4,4,5,6,6,7,7,7,7,8] [100,90,80,70,60,50,40,30,20,10]% contrast components); Chirp: 47 components; SK: 8 components in total (2 R components, 2 G components, 2 C components and 2 B components), giving a grand total of 140 PCA components.

We clustered the combined 'scores' matrix using Gaussian Mixture Model (GMM) clustering, performed using the Matlab routine **fitgmdist**. We initially sorted the data into 100 clusters using (i) shared-diagonal, (ii) unshared-diagonal, (iii) shared-full and (iv) unshared-full covariance matrices, such that four different clustering options were explored in total. For each clustering option 20 replicates were calculated (each with a different set of initial values) and the replicate with the largest loglikelihood chosen. A regularisation value of $10^{-5}$ was chosen to ensure that the estimated covariance matrices were positive definite, while the maximum number of iterations was set at $10^4$. All other **fitgmdist** settings were set to their default values.

The optimum clustering was judged to be that which minimised the Bayesian information criterion (BIC), which balances the explanatory power of the model (loglikelihood) with model complexity (number of parameters). Using the above procedure, we found that unshared diagonal covariance matrices provided the optimal solution (i.e. the solution with the lowest BIC). From here we combined 100 initial clusters into 36 merged clusters by hand based on similarity in cluster mean responses. Finally, of the 36 remaining "compound-clusters", 14 that comprised fewer than 20 members were discarded, yielding a final total of 22. The full dataset, including allocations to the original 100 clusters and their subsequent combinations are available online at http://chicken-data.retinal-functomics.net/.

### Quality indices

We computed three separate response quality indices pertaining to the WS and CS stimuli as well as the spectral kernels. The quality indices for

WS and CS stimuli were calculated as:

$$QI = (max(SYNC) - mean(ISI) + (max(PSTH) - mean(PSTH)) \times std(SYNC)$$

(3)

where SYNC denotes the SPIKE-synchronization, ISI the inter spike interval and PSTH being the peri stimulus histogram calculated over all 5 repeats of stimulus repetition. SYNC, ISI and PSTH were calculated using the Pyspike Python package[88]. Kernel quality was calculated by adding the sum of the 100 highest counts and the absolute sum of the 100 lowest counts of z-normalised STAs across all chromatic channels. Based on these, we then computed a 'compound QI' for each cell by first normalising the distributions of each of the three individual QIs (CS, WS, kernels) to a mean of 1, followed by averaging across the three normalised QIs. As such, the three stimuli were given equal weighting. This compound QI was used as the basis of sorting cell by quality within each cluster. Note that a different set of QIs was used to preselect cells for inclusion in the clustering in the first place (see above).

### Step-responses (WS, CS)

*Response amplitudes.* We computed three basic response amplitude measures for each step-transition. (i) The total On and Off response amplitudes were taken as the mean number of spikes elicited during the full 2 s of stimulus presentation. These are used as the basis of all plots showing spectral tunings and contrast response functions (e.g. Fig. 4j, k), to compute the polarity index (e.g. Fig. 3b) and to compute the spectral tuning index (e.g. Fig. 5j). We also computed a baseline value per cell as follows: A 1,000-bin amplitude histogram was computed based on the concatenated, cluster-averaged responses to WS and CS, and the peak of this histogram was taken as the baseline. Accordingly, the baseline was defined as the most common amplitude value throughout the full response trace to these two stimuli, which yielded reasonable estimates as judged by manual inspection. This cluster-wise baseline estimate was also individually associated with each cell within a cluster because this procedure was judged to yield more robust cell-wise estimates compared to computing the same metric based on each cell. The baseline value was used in two ways: As a display item in the spectral tuning and contrast-response plots (e.g. Fig. 4j, k), and as a means to normalise response amplitudes across clusters, where tuning functions were normalised between 0 (baseline) and 1 (peak response, e.g. Fig. 4l, m).

For each step transition (i.e. On and Off), we also computed 'transient' and 'sustained' response measured based on the peak spike rate within time windows of 80–160 and 240–2000 ms following the step transition, respectively. These time windows were chosen as a compromise to capture response properties across all analysed species' datasets despite their somewhat different overall kinetics. The exact choice of these two time-windows did not qualitatively affect the results. To ameliorate the effects of noise, we used box-smoothed (window size of 40 ms) response traces to estimate these latter two metrics. The transient and sustained response metrics were used as the basis of the transience indices (e.g. Fig. 3b). The transient-response amplitude was further used as the basis of spectral dominance index (Fig. 5j, k, Supplemental Fig. 6), and to compute normalised amplitudes used for relating latencies and amplitudes in Fig. 6e, f.

*Polarity index (PI).* For clusters, the polarity index (PI) was computed based on On- and Off-response amplitudes (see above) of the 100% contrast WS as:

$$PI = \frac{A_{On} - A_{Off}}{A_{On} + A_{Off}}$$

(4)

Where $A_{On}$ and $A_{Off}$ are the On- and Off-response amplitudes, respectively. Accordingly, PI ranged from −1 to 1 to denote entirely

Off- and On-dominated responses, respectively. A PI of 0 denotes a cell with equal amplitude On- and Off-responses. The same measure was also used to compute polarity of the zebrafish, mouse and human datasets (Fig. 3d–f, Supplemental Fig. 4d), with the exception that in this case, $A_{On}$ and $A_{Off}$ were taken as the total response for 1 s (rather than 2 s) following a step-transition. This adjustment was necessary because the step-duration in the mouse dataset was 1 s.

*Transience index (TI).* As for polarity (above), the transience index (TI) was computed as the contrast between transient and sustained responses per 100% contrast WS as follows:

$$TI = \frac{A_{Tr} - A_{Sus}}{A_{Tr} + A_{Sus}}$$

(5)

Where $A_{Tr}$ and $A_{sus}$ are the transient and sustained response amplitudes as described above. Accordingly, TI ranged from −1 to 1 to refer to cells entirely dominated by their sustained and transient response component, respectively.

For each dataset, the TI was computed three times: As On- and Off-versions, based on the respective On- and Off- measures of transient and sustained responses, and as a 'compound TI' where the associated polarity index would dictate which of the On- or Off-versions would be used for transience. This 'compound TI' was used for Fig. 3b–e, while the individual On- and Off- TIs were used for Supplemental Fig. 4e.

*Spectral dominance (SD).* Spectral dominance (SD) was computed as follows:

$$SD = \frac{A_{colour} - A_{white}}{A_{colour} + A_{white}}$$

(6)

where $A_{colour}$ and $A_{white}$ are the transient response measures of the largest CS responses, and that of the 100% contrast white response. SD ranged from −1 to 1, indicating responses entirely dominated by WS and CS stimuli, respectively.

### Spectral tuning (ST)

An index of "spectral tuning" (ST) was devised to indicate how closely the spectral tuning of a given response matches that of any of the four opsins expressed across the chick's cones (LWS, RH2, SWS2, SWS1[20]). To compute ST, we first log-transformed Govadovski-templates[89] of each of the four opsins and stretched it over one log-unit, such that in each case a linear response of 10% and 100% maximum was mapped to zero and 1, respectively. Such a -10-fold log-transform is expected based on the phototransduction cascade of cones[90]. The resultant transforms are also used as the basis of all opsin-templates presented across the figures. For each cell or cluster's normalised spectral tuning function (see above) we next determined which of the four templates provided the closest match based on their correlation coefficient. We then subtracted the normalised tuning functions from their respective template, and computed ST as the mean of their difference, multiplied by −1. As such, ST was zero when the opsin template and response were perfectly matched, but negative and positive, respectively, if the response was spectrally broader or narrower than the opsin. Throughout, we applied a minimum response threshold of 3 spikes as the peak response—tuning functions based on fewer spikes were not considered.

### Latency

Response latency was computed separately from On- and Off-transitions of all time-smoothed WS and CS responses (40 ms window size) as the time to half peak.

### Principal component analysis (PCA)

To compare the temporal response-envelopes elicited by CS and WS stimuli, we used principal component analysis. Computing separately for On- and Off-transitions, for a given cluster (Fig. 6i, j), or the mean responses of all OnOff clusters (Fig. 6g, h), we combined the first five CS (i.e. excluding UV)

and first five WS (100, 90, 80, 70, 60% contrast) into a 50×10 input matrix (50 time bins of 20 ms each, 10 responses). To exclude residual activity from the preceding stimulus, we zeroed the first 4 bins (=80 ms). The first two principal components emerging from PCA across this matrix consistently explained >94% of the total variance. Accordingly, higher PCs were discarded. To relate the results from the PCA across clusters (Fig. 6i, j), sets of loadings were individually peak normalised to 1.

## Spectral kernels (SK)

**Kernel amplitudes.** Kernels were individually (R, G, C, B) z-normalised based on timepoints between 1000 and 500 ms preceding the spike, and amplitudes were subsequently computed (in z-scores) as the difference between their maximum and minimum values. Cluster-mean kernels with an amplitude <2.5 were discarded from further analysis.

**Spectral centroids.** To quantify the kinetics of each kernel, we estimated their central frequency in the Fourier domain ('spectral centroid') as follows. We first computed each (R, G, C, B) kernels' probability mass function as its area-normalised magnitude-squared Fourier transform. We then multiplied each entry by the reciprocal of its associated frequency in Hz. The kernels' central tendency in Hz was then taken as the sum of this product.

**Colour opponency.** Colour opponency was determined based on the relative trajectories of each cell or cluster's group of four spectral kernels (R, G, C, B). Due to the complex interplay between time and polarity across the group of kernels, we started by categorising each kernel-group's 1 ms time-bin as either non-responsive, responsive and non-opponent, or responsive and opponent. Time-bins where none of the four kernels exceeded a minimum absolute amplitude of 3 (or 1.5 for cluster means) were defined as non-responsive and set to NaN. All remaining responsive bins were categorised as either opponent (1) or non-opponent (0) depending on the relative signs of all four kernels amplitudes. Next, for all kernel-groups where at least 20 (of 1000) time bins were categorised as opponent (i.e. 20 out of 1000 ms maximal kernel duration), a colour opponency index (COI) was defined as the mean of this vector. Accordingly, colour opponency ranged from 0 (entirely non-opponent) to 1 (entirely opponent). Kernel-groups with a Kernel-QI (see above) <1000 were excluded from this analysis and categorised as non-opponent. The time-dependent opponency vectors were further used as the basis of aligning the opponent and non-opponent time-regions of each OnOff cluster (Fig. 6a, b). To this end, we time-compressed each associated vector between its beginning (time −1), defined as the first timepoint where the group of kernels exceeded a minimum amplitude of 1, and the time of the spike (time 0). In this case, non-opponent kernel bins were set to −1 instead of 0 (Fig. 6a, b)

## Chirps

**Fourier analysis, best frequency, and high-frequency index (HFI).** For each cluster mean, we computed the magnitude-squared Fourier transform for the full chirp-response (accelerating from 1−30 Hz over 30 s−shown as area normalised in Fig. 4e, f), and used this as the basis of a high-frequency index (HFI):

$$HFI = \frac{P_{high} - P_{low}}{P_{high} + P_{low}} \qquad (7)$$

where $P_{high}$ and $P_{low}$ were the mean power between 0.9 and 2 Hz, and between 4 and 15 Hz, respectively. As such, HFI varied between −1 and 1 to indicate clusters entirely dominated by a low- and high-frequency response, respectively. Clusters that exhibited fewer than 20% additional spikes during the chirp-stimulus compared to the 5 s

preceding it were considered unresponsive and excluded from this analysis.

Best frequency was taken as the peak of each cluster's magnitude-squared Fourier transform.

**Phaselocking analysis.** For each spike elicited during chirp stimulation, we computed the phase relative to the stimulus as follows. First, to compensate for the phototransduction delay and inner retinal processing, all spikes were negatively time-shifted by 100 ms relative to the stimulus. The value of 100 ms was chosen based on the mean of On- and Off-latencies (time at half maximum) across all cells' 100% contrast WS-responses (=99.4 ms). We next divided the 30 s chirp segment (1−30 Hz) into ten equal time bins of 3 s each, with their corresponding mean frequencies centred at 1.2, 1.7, 2.4, 3.3, 4.7, 6.5, 9.8, 12.5, 22.5 and 25.0 Hz. We then computed the phase angle θ for each spike (ranging from 0 to T, where 0 and T/2 denote the onset of the On and Off-phase, respectively, as indicated e.g. in Fig. 3h). For each cell and time bin we then determined resultant median vector strength r (ranging from 0 to 1 to indicate a random relationship between each spike and the stimulus, and perfect phase locking, respectively) as described elsewhere[91]. For statistical testing, we also computed the same metrics based on 1,000 Poisson statistics but time-randomised artificial neurons that comprised the same number of average spikes as each cluster. We then used the upper 95% confidence intervals from the distribution of vector strengths emerging from this artificial set of neurons as the basis of determining statistical significance (i.e. $p < 0.05$): Any cluster-entries that failed to exceed this confidence boundary of their corresponding bootstrapped distribution were deemed non-significant. For clarity, these data points are removed from the presented data (e.g. Fig. 4h).

**Choice and pre-processing of functional RGC datasets other vertebrate species..** To relate our data from chick retina to those of other species, we sourced previously published RGC-response datasets from human, mice, and zebrafish. To ensure good compatibility, we specifically looked for step-responses to spectrally broad stimulation, and for datasets that comprised representative fraction of most, if not all RGC types in that species. However, since the state-of-the-art with regards to different species is not fully equivalent, some compromises had to be made. The mouse dataset was sourced from ref. 43 (rgc-types.org) as probably the most comprehensive available to date. The dataset presents mean spike-responses of all 42 RGC types currently described in mice, to a 1-second step of scotopic light aimed to stimulate the rod-system. The relative abundance of each type was then complemented by assessing their corresponding abundance on Eyewire[49]. Next, as an example for primates, we chose a multielectrode array dataset collected from post-mortem human donors, published as part of ref. 44. While our understanding of the functional RGC types in the human retina is perhaps still lagging that of some other primate species, we nevertheless chose this dataset because it included systematically recorded 'white' step responses−in this case for 3-second 100% contrast low-photopic steps of light. The dataset consisted of $n = 411$ individual neurons. From here, we clustered the data using Kmeans into 20 clusters that capture some of the major structure. We chose the value of 20 as a compromise between a very conservative estimate of the full RGC diversity in primate retina, and in view of the limited dataset which is likely to lack some of the rarer types. The exact number of clusters had no notable effect on the overall distribution of clusters with regards to polarity and transience (see also the original publication[44] which shows a very similar distribution based on 5 clusters). Finally, we also included our previously published functional RGC dataset from larval zebrafish[10], which included 3-second step responses to 100% contrast low-photopic white light. Because this data was collected by 2-photon mGCaMP6f imaging of RGC-responses in the live eye, rather than by recording ex-vivo RGC spikes, we

approximated spiking activity by deconvolving responses with a Tau = 0.4 s decaying time kernel. The exact value of this deconvolution did not notably affect results. A correspondingly comprehensive zebrafish dataset based on direct measurements of spike responses has not been published. Because compared to mice and primates, in larval zebrafish the link between functional[10], anatomical[71] and genetic[92] RGC types remains notably less complete, we opted to use the n = 16 functional RGC clusters and their relative abundances that were described in the original study as the basis of our analysis. It is likely that this relatively low number of clusters underestimates the full RGC diversity in larval zebrafish. Since the duration of step responses used across the 4 datasets were not identical, we cropped the first 1 s following each On- and Off-transition and resampled each into fifty time-bins of 20 ms. From here, analysis for all datasets was identical, and followed the steps for computing polarity and transience indices described above. When separately computing histograms of On- and Off-transience, applying a minimum response threshold of 20% for each type/cluster's 'weaker' polarity for inclusion, where applicable (such that e.g. a cluster that exhibited a more than 5-fold stronger response to On versus Off was included for the On-histogram, but not the Off).

## Quantification and statistical analysis

### Statistics

No statistical methods were used to predetermine sample size. Owing to the exploratory nature of our study, we did not use randomization or blinding.

Linear correlation tests were used to establish the correlation coefficient and significance of relationships between polarity and transience indices (Fig. 3a, d–f).

Wilcoxon Rank Sum tests were used to probe for differences in conduction velocity between different cluster groups (Fig. 4i). The custom bootstrapping approach associated with evaluating vector strengths (Fig. 4g) is detailed in the section on phase-locking.

### Reporting summary

Further information on research design is available in the Nature Portfolio Reporting Summary linked to this article.

## Data availability

The processed (spike-sorted) RGC responses of each cell included in this study are available at https://sussex.app.box.com/s/2emxshif961gdjx4ht63lfxplazdosfo, in different formats, including download instructions. The raw MEA data are available upon request, due to their immense size. Source data are provided with this paper.

## Code availability

All scripts for non-trivial analysis steps (electrical imaging, reverse correlation of spike times etc.) are available on GitHub https://github.com/BadenLab/Nature_Comms_paper_23.git. We included a requirements.txt listing all python dependencies as supplementary data.

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

## Acknowledgements

We thank Thomas Euler for critical feedback. The authors would also like to acknowledge support from the FENS-Kavli Network of Excellence and the EMBO YIP. Funding was provided by the Wellcome Trust (Investigator Award in Science 220277/Z20/Z to T.B.), the European Research Council (ERC-StG "NeuroVisEco" 677687 to T.B.), UKRI (BBSRC, BB/R014817/1 and BB/W013509/1 to T.B.), the Leverhulme Trust (PLP-2017-005 and RPG-2021-026 to T.B.; G.K. was supported by the Leverhulme DTP DS-2017-011) and the Lister Institute for Preventive Medicine (to T.B.). This research was funded in part by the Wellcome Trust [220277/Z20/Z]. For the purpose of Open Access, the authors have applied a CC BY public copyright licence to any Author Accepted Manuscript version arising from this submission.

## Author contributions

Conceptualization, M.S., D.O., T.B.; Methodology, M.S., P.A.R., G.K.; Investigation, M.S., T.B.; Data curation, M.S., P.A.R.; Writing – Original draft, T.B.; Writing – Review and editing, M.S., P.A.R., G.K., D.O., T.B.; Visualization, M.S., T.B.; Supervision, D.O., T.B.; Project administration, D.O., T.B.; Funding acquisition, D.O., T.B.

## Competing interests

The authors declare no competing interests.
