## [Peer Review File · Nature Communications]

Birds multiplex spectral and temporal visual information via retinal On- and Off-channelsREVIEWER COMMENTS

Reviewer #1 (Remarks to the Author):

The study “Birds multiplex spectral and temporal visual information via retinal On- and Off-channels” investigates how retinal ganglion cells (RGC) in birds (poultry chicks) represent visual information. The study aims at answering whether principles discovered in the mammalian retinal circuits also apply to the avian retina. To address this very interesting question, the authors established an ex-vivo approach for recording light-induced neuronal activity from retinal ganglion cells in poultry chicks using a high-density multielectrode array. To characterize the response properties of the retinal neurons the authors used a rich and established stimulus set which allowed the authors to identify functional RGC groups within their dataset.

The novel and the key finding of the study is that RGCs in chicks represent visual information in a different way compared to mammalian RGCs. While information about the polarity and kinetics of visual signals are represented in a decorrelated manner in mammalian RGCs, this study now reports that these properties are represented in a correlated manner in chicks. This finding is surprising and exciting because it challenges the long-standing view that visual information is efficiently represented by decorrelated visual channels. Moreover, the results question the assumed common architecture of all vertebrate retinal circuits. The presented results are thus of clear relevance for the discussion in the field and will have a significant impact on it.

I very much enjoyed reading the manuscript and the presented data look solid and of high quality. However, I do have several major comments in regard to the presented results and the conclusion that the authors draw from them.

1) The authors report that most RGCs encode both ON and OFF signals, with ON signals being more sustained and OFF signals being more transient. This is in contrast to what has been reported in mammals and thus a key result of this study.

This conclusion is based on extracellularly recorded neuronal activity and the authors used a spike-sorting approach (Herding Spike2) to assign action potentials to single clusters/neurons. While spike-sorting is a standard and integral part of extracellular recording methods, it works, unfortunately, never perfectly and can and will lead to some level of mixing of multiple single neurons into single clusters. Thus, potentially a OnOff RGC neuron with slow ON and fast OFF kinetics could be the result of combining the spikes from an ON-slow RGC and an OFF-fast RGCs into a single cluster. Therefore, a key conclusion of the study could potentially be affected by imperfect data analysis that results in a mixing of neuronal signals. I do not think that this is necessarily the case, however, in the current form, the manuscript contains very little information on the spike-sorting results that would allow the reader to assess the spike-sorting quality and to convincingly show that there is no systematic bias/error introduced by the spike-sorting.

Therefore, please provide more information on the spike-sorting quality for all clusters, i.e. unit quality metrics and the inclusion criteria etc. Moreover, please also test whether neurons from different functional groups show some sort of systematic bias in regard to the spike-sorting quality. For example, it would be informative to analyze whether the OnOff polarity index is correlated with

the spike amplitude or similar quality metric, i.e. whether the OnOff neurons are predominantly found in clusters with small spike amplitudes/clusters with lower spike-sorting quality. Again, I am not saying that this should be the case, but it is important to rule out any potential contamination of the results by the spike-sorting part of the analysis.

Related, the authors write: “We computed the “electrical image” (EI) for a subset of spike-sorted cells)” and state that this image permits to infer important information about the neuron type, which I agree it does. However, it is unclear how the neuron type was inferred for neurons in which the electrical image was not computed. Therefore, please specify the fraction of neurons for which the electrical image was not available and how the neuron type was inferred in those cases.

2) The functional clustering of RGC types is another important analysis. While the clustering analysis is described in detail in the method section and briefly mentioned in the results section, important results are not shown in the manuscript which makes assessing the quality of the clustering difficult. For example, the Bayesian information criterion was used to judge the optimal number of clusters but the results are not shown anywhere in the figures. Please provide more information on the clustering and grouping within the manuscript figures or supplementary figures.

3) Comparison to other vertebrate species. Although not statistically significant, the human data appears to follow a similar trend as the fish data (Figure 2d-f). Is the outlier data point in the lower left quadrant in Figure 2f responsible for the statistical insignificance? If that is the case, please further assess the robustness of the results.

Please also provide the p-values from the correlations shown in Figure 2d-f such that the reader has access to this important information.

Related, why are the values for the transience index below 0 for many neurons in the chick but almost all above 0 for fish, humans and mice?

4) The data appears to be from the dorsal retina and not from both the dorsal and ventral retina regions. Given that several studies have recently reported differences between dorsal and ventral retinal in mammals I was wondering whether the presented results in the chick retina generalize across the dorsal and ventral regions. Fully answering this question is of course beyond the scope of this study, but it would still be informative to discuss the presented data in this context.

Minor comments:

- Figure 1n: Why are the responses of only 5 out of the 6 wavelengths shown?

- Figure 1o: Why is the chirp stimulus strange at high-frequencies? Is this just a rendering issue with the figure or was the stimulus like that?

- Extended Data 1e: please provide the real number of cells in addition to the circles. While the different-sized circles are a nice visualization, proper numbers would help to better understand how many cells were included in each group.

- Figure 3d: The authors write that the flicker responses for C1,2 were ON type at low and OFF type

at high frequencies. However, Figure 3d suggests that, at least at low frequencies, the responses happen during the downward phase of the chirp, which should be the OFF part of the response. Should the responses not be at the positive part of the chirp at low frequencies if?

Reviewer #2 (Remarks to the Author):

This manuscript reports and provides interpretation of a large dataset of retinal ganglion cell recordings in the retina of chicken chicks. These are absolutely novel data that add considerably to our current knowledge on the retinal computations that take place in the retina of birds. New findings include – among others - the dominance of on-off channels in the bird retina, as compared to the separation of separate on and off channels in the mammal retina, and a suggestion how both chromatic and achromatic information may be coded in the bird retina. Already the large data set reported here is worth publication, but the authors go beyond this and compare their results with those from other vertebrates. While some of the interpretation is maybe speculative, such parts are clearly marked, and the conclusions are well founded in the results. Methods are complex but mostly well described, and the development of a methods allowing reliable recordings from bird retina is a major milestone also for future work on bird retinal coding.

Clearly this paper should be published, as it adds important new findings on a long ununder-studied system. My two minor comments concern (1) missing details on methods in the main part of the manuscript (which make it more difficult to understand for a general reader), and (2) the perspective that seems to largely take a mammal-centric perspective, while evolutionarily, birds are likely more similar to an ancestral state of vertebrate retina, as they have not lost two cone types. This evolutionary interpretation is given in the end, but it would be more convincing to start with it, and treat the mammal situation as derived from an ancestral state rather than “the state of knowledge” from the start.

Else, the paper, despite long, reads well, while the figures, specifically the first figure are crowded making it sometimes difficult to really appreciate the data. If possible, making two figures from figure 1 (with larger part figures) would probably improve the paper.

All of these points are minor.

Detailed comments

The abstract: it is written as if for a specialist journal – it may be good to start with a more general sentence clarifying that the paper is on vision. For instance, there are animals other than vertebrates that have visual systems, but the first sentence is written as if it was general, and then only in relation to other vertebrates. The word “bird” appears in thesecond- last sentence only. Keep the general reader in mind for the abstract

Line 13: an English sentence cannot start with “But”, please reword

Highlights: I am not convinced by the first one. What is “visual function in an avian retina”? Could it be more telling and specific, maybe First large scale survey of retinal ganglion cell function in an avian retina. One could argue that cone sensitivity is also a “visual function in the avian retina”, and

this has been studied for a long time.

Introduction:

Like the abstract, this is written for specialists, it may be difficult for a general reader to understand*.

Line 73: while the zebrafish work from the Baden lab is amazing and admirable, they were not the first to study fish retinal circuits, and it would be fair to mention some older work, at least in passing.

Line 76: "retinal functions" is a very broad term, which – in my understanding - includes everything from oxygen supply to visual pigment turnover, may be better to say what you mean: the coding of information by the inner retinal layers. It may be helpful to mention for a general reader which cells a retina has and which fulfill the functions mentioned above and below.

Another important piece of missing information – at least for a general reader but maybe even for primate vision researchers, is the evolutionary aspect. This entire data set and its comparison to mice, primates and fish, is most interesting from an evolutionary perspective. As you are aware, while fish may be closest to the ancestral visual system (even though bony fish may have moved quite a bit from that situation), birds are more similar to them as they have kept the full complement of visual pigments and thus, receptor types (even though, it may be debated that double cones are new, compared to fish), while mammals have lost two cone types and likely, with them, retinal neuron types. So chickens are much more general than any mammal, and primates even have a visual system that uses a spatial coding channel to code colour, which makes them the least general of all, and all difference to them may be quite expected. Salamanders and turtles are likely similar, but the best reason to study birds and not these in detail, is that the birds are the absolute vision-specialists among vertebrates. I think there needs a short mention here to prepare the reader. Otherwise, the reader may ask, why do we need to know anything about chicken in the first place?

*For instance, line 90: "chicks represent information about the polarity, kinetics, and wavelength". First, say chicken (they also have a Latin name which needs to be mentioned at first mention) chick means just a young bird, you studied chicken chicks. You need to first say the full term, then you can continue with just "chicks". Second, it is very colloquial to say that chicks represent... - the signals recorded from cells in the RGL represent the properties of visual stimuli. Visual stimuli or light stimuli needs to be mentioned in the sentence, else it makes no sense. Sorry, colloquial language can make reading difficult. At this point, the reader has not yet been told what on- and off-circuits are.

Results

Line 130: you use "n" for both the number of preparations and retinae – would it be better to use "n" and "N"? It will be helpful to mention the size of the retinal pieces as the methods section is far away.

Line 134: displaced amacrine cells: first mention here. I really think the retinal cells that will be relevant for understanding the results need to be mentioned in the introduction.

Line 136: just a side note: I really like the presentation of these 'electrical images', very helpful! In the figure, is it possible to indicate some information on the orientation of the cells in the retina (what is up and down, right and left, as you mention in line 140 that the axons of RGCs should be oriented towards the optic disk – how would you know?). Would it make sense to also show an example of a displaced amacrine cell, maybe in the supplement?

Line 155ff: this is general information on the chicken retina, which I would have placed in the introduction. I find it strange to place it here.

Lines 167ff: even though stimulus steps are given in the methods, can they be repeated here (n photons or photons per s or something like that, in parentheses)? More than that, what does “corresponding” mean for the colour steps – same number of photons, or same number in the wavelength range? Important to know here without having to search in the Methods.

line 204: quite frustrating not to get any idea of what “Mixture of Gaussian Model” means unless I move to the Methods part. Is there a short way to say it here?

Line 225: “This revealed surprising trends” is a strange way to start a description, as the reader is not aware of the authors’ expectation and thus, would not know what should surprise them. Is it possible to first describe the results and then why they are surprising? Moreover, in which sense is the word “trend” used here? Often, it is used for “just not significant” results, something I dislike, so please clarify how the word is used here.

Line 226: I guess cells responded to ON and OFF. Clusters have earlier been described as responses, and responses do not respond. And certainly, clusters don’t respond. Cells grouped in clusters respond. Thanks for clarifying the writing. What is a “large-amplitude kernel, in terms of cell responses? Does it mean that cells responded with high spike frequencies, or to high amplitudes of stimulus intensity? I generally think it is more correct to say “high amplitude”, not “large amplitude”.

Line 287ff: the comparison is rather superficial. Neither are potential reasons mentioned nor how the groups are similar or different. I would add both either here or later. A major reason for differences is the fact that mammals lost two visual pigments and thus, two types of cone, and thus, may have lost horizontal, amacrine and ganglion cell types.

Line 295: I have no clue what – quantitatively – a “sizeable complement” is meant to say. Can you please give the reader less cloudy wording, and, preferentially, also numbers?

Lines 414ff: I think the two indices are a very useful way to describe the data. Therefore, I am wondering whether these have been newly invented for this paper, or are used elsewhere – no references are given, so I guess they are new. If that is correct, it would be interesting to know whether there is any way to compare the results to those obtained from other vertebrates (as was done with other aspects in earlier sections).

Line 546: “For of the Off-channel, this trend extended to blue” – remove the “of” here.

Line 553ff: I am not sure about the comparisons between the different colour channels; what do they say given that unfiltered red cones have broad sensitivities, and that the relative intensities of the four coloured stimuli are unlikely to be matched for any of the receptor types. They may be descriptive of the data here but may not tell much about the chicken retina, so it may be good to be more cautious here? For instance, a response to white light may be more similar to a response to red light, as the majority of cones in the retina (double cones plus red cones) are red-sensitive. The sensitivity curves of unfiltered cones overlap considerably, making differentiation less clear than it will be in the real retina with oil droplet filtering. These points may have to be pointed out when interpreting results.

Line 554: The “red response” is later (line 569ff) used as reference, so it may be important to clearly define what it is – the response to the highest step of red light? Please define here to help the reader understand exactly what was done. The other responses, I assume, are defined accordingly, which also needs to be said.

Line 587: unclear why, after “all...”, the word “step” is singular, please check the sentence, moreover, in the next line, please add “white” if you refer to white-light steps there, else please specify.

Line 584: not sure I have seen a definition of “temporal” envelope. While it is likely clear to many readers, it may not be clear to all,

Line 594: while clearly, the two types of analysis (amplitudes and latencies versus PCA) are different, the PCA also uses amplitude and latency information, and thus, results are not independent. As is written, it is not clear whether a PCA of just amplitude and latency would give the same result, and thus, which contribution the “temporal envelope” makes.

Line 617: in a results section, I would not speculate on “perhaps also in some other non-mammalian species” – Maybe you could generalize “and perhaps also in other birds” but anything broader makes little sense here, and seems to reflect a very narrow mammal-centric perspective.

Discussion

The discussion also starts mammal-centric. Not sure this is meaningful unless the authors submit the paper to a journal mostly read by mammal vision scientists.

Line 674 f: this is an odd sentence, maybe reword? The entire paragraph is speculative (nothing has been learned about space, in the present study so why speculate?) anyway and could also be removed.

Line 741: I thought frogs also have double cones (e.g. Nilsson 1964 but also Donner and Yovanovich 2020) – are you sure only birds and reptiles have them?~ Only birds have double cones that always express LWS in both members, reptiles as well as frogs have different combinations of opsins expressed in both members. And while they are independently wired, there are good data indicating both members are connected by gap junctions, maybe worth mentioning as it must have functional implications.

Line 767ff: very interesting paragraph, just an additional thought: Maybe, LWS off is a very good reference, as it is a highly sensitive channel? In addition, this may be a reliable way to make the chromatic channels independent of intensity.

Line 785: equally interesting but “cortex-like” is not convincing – reminds me of a very naïve interpretation of stomatopod photoreceptor tuning being cortex-like. It is always good with the most simple explanations, unless they can be disproven, and nowadays, there is a lot of evidence (from invertebrates as from vertebrates) that already at the first synapse in the retina, the spectral signal may be influenced by opponent input from other spectral types, in vertebrates mediated by horizontal cells. If this is what you think happens, it would be good to mention it and give references.

Methods

Animals: can you give the age at which chicks were sacrificed, please?

Light stimulation: it is important to give at least some intensity measure of the used light stimuli. It may be useful to express intensity in terms of the number of photons per time unit and visual angle and area reaching the level at which the retina was positioned. It is also important to know how the intensity of the single LED stimuli related to the intensity of the white light stimulation.

Line 1077ff: while it sounds likely that the selection of methods was sound, saying "we found that unshared diagonal covariance matrices provided the optimal solution" may leave the reader wondering how "optimal" was decided upon. Is it possible to clarify? A methods reference may also help.

Line 1108ff: similarly, is there any way to explain whether you expect a different result in any way, if you modify the analysis but, for instance, changing time basis (eg only taking the first 1s after stimulus on or off)? Robustness of results is important.

Figure 1 (and general) vision scientists may be aware that there are colour-blind and colour-deficient readers who need to understand the figures as well, so in addition to colour-coding stimuli, please add a letter somewhere to explain the stimulation wavelength. Please check for all figures. Interestingly, the occurrence of colour-deficiencies seems to be higher in vision scientists than in the general population.

More generally, I find figures very crowded and wonder whether authors are forced to combine so many part figures in days of electronic journals, for which space should be less of a cost. Some part figures – important ones such as 1m-1p) are so small that readers will miss a lot of important and interesting details. If possible, can you give these an own figure and make it a little larger (not all of them in one row)?

Figure 2: d-f: it is said "as b" but the clusters are all grey – so are there only sustained ON responses shown? Else, please add that there is no colour code here. Please also explain the lines in d-f.

Reviewer #3 (Remarks to the Author):

Review of NCOMMS-22-49472, 'Birds multiplex spectral and temporal... '

This is an interesting study of population-level visual encoding by ganglion cells in the chick retina. The authors developed an approach to record chick retina on a high-density multi-electrode array (MEA) and studies the responses of all isolated units across the recorded area to a stimulus with broad spectral and temporal content. This approach mimics that of earlier, highly cited work by some of the same author in mouse, and allows reasonable comparison between results between an avian and a mammalian species. The manuscript has a strong comparative component, which is both appropriate, and timely considering the current acceleration of knowledge of mouse retina (mammalian) where, despite initial discoveries and decades of research, amphibian, avian, and fish retina have not entered this somewhat new era of comprehensive functional study (with some exceptions).

The authors reasonably use cluster analysis to group isolated units. The number of units is less than apparent number of ganglion cell types reported from gene expression study, so that clusters may contain multiple types with similar response properties – acknowledged by the authors. The authors report that ON-OFF response types predominate, and broad groups – ON, OFF, and ON-OFF – differ in their temporal response tuning and encoding, as well as in their spectral tuning: OFF response are fast and largely achromatic, ON responses are slower and carry the bulk of the chromatic information. ON-OFF response types carry both signals, and this is reasonably interpreted as multiplexed encoding, where the early, fast OFF component can be decoded separately from the later spectral ON component. Comparison across clades is extensive and well done and the overall volume of new information about chick retina function considerable. I find this a solid study that helps broaden our current fairly (dangerously?) narrow, mouse-centered understanding of retinal functional organization.

I have the following comments.

1. Multielectrode arrays have a recording bias. Unless I missed it, this, and its possible impact on the results (number of clusters? Over-representation of some types) is not addressed in the manuscript, but some explanation of the possible impact would be appropriate.
2. Related, MEA ‘shows’ cells in a recorded area. Morphology would, too. What % of the morphologically identified ganglion cells is recorded on the MEA?
3. The authors use reverse-correlation analysis to measure visual response kernels for the recorded units. Reverse-correlation measures linear response properties, whereas ON-OFF responses, for example, are obligatory nonlinear. How is this dealt with in the analysis, and what may be missed?
4. A major issue with the results as presented is that most analysis is performed on cluster averages, i.e., the average response waveform of all units assigned to a cluster (line 218). Because of this simplification, variance on the responses within each cluster is gone, which obviates statistical comparison of response features across clusters. This is a major problem that impacts many claims, beginning with the results of Figure 2, and must be addressed. The results are (cleverly?) introduced as ‘surprising trends’ (line 225) but are presented as more than ‘just’ trends throughout. Please address.
5. Line 63: a study by Ratliff et al., PNAS 2010 (with additional relevant references in Introduction) shows that the division of visual encoding in the retina is distinctly not equal. Consider qualifying this statement.
6. Figure 1 and elsewhere: the temporally accelerating (chirp) stimulus waveform (panel O, bottom) shows distinct aliasing of the stimulus, with basically ON-OFF flicker just prior to 5s, and near-constant ON around 20(?)Hz (above the 2 s label). Does this accurately depict how the stimulus waveform was mapped onto the stimulus monitor/light source? Please address this in figure legend or Methods, and explain how this impacted the stimulus present at the photoreceptors.
7. Line 151: references non-myelinated axons. Aren’t all axons in the retina non-myelinated? My understanding is that myelination begins at the optic nerve head.
8. Line 243: It is shown here that 19.7% of recorded cells were ON, *transient*, non-opponent. The ON sustained fraction is much smaller, just 6.6%. How does this rhyme with the statement on line

229 that 'ON-dominated clusters were sustained'? This is an important discrepancy, as the entire thesis of the manuscript is based on ON = sustained.

8. Figure 2b: (1) consider making explicit here or in the text what is negative transience. If positive is transient, is zero sustained, and negative a temporally increasing response? (2) if orange symbols are transient and gray symbols sustained, then how can there be gray clusters and orange clusters at the same (negative) transience level?

9. This is a comment: Fig2D, E, F – nice that data sets from three different vertebrate species are shown here for comparison.

10. Line 269: typo, as Figure *2*a.

11. Line 432-433: It is counterintuitive that drive from a single opsin as proposed here for OFF cells would give broad, achromatic tuning. Would their tuning not reflect that of the opsin? Perhaps briefly explain here how this works.

12. Line 672: ra*n*ge.

General preamble for all reviewers

We thank all reviewers for taking the time to carefully review our work, and of the many useful suggestions made. As detailed in the below, we have tried, where possible, to follow all these suggestions. This resulted in some major changes in the presentation of the MS, in particular in view of some methods aspects regarding spike sorting and clustering. Importantly, the big picture that emerges is that despite the additional tests now included, all major conclusions from the previous version of the MS stand up to this additional scrutiny.

Major changes:

- Extensive additional testing of spike-sorting quality, and if any errors here could have influence the results in a systematic way. Overall, we find no evidence that the OnOff behaviour of many units (a core results of the study), are in any way linkable to systematic errors in spike sorting. This is now discussed at length in the methods section and summarised in new Supplemental Figure S2.
- New additional consideration of how variations in clustering (and spike sorting) could have led to errors in data interpretation. This is now explicitly discussed at the beginning of the results section when first introducing the clusters. As above, the central conclusion is that the specific methods of clustering did not drive any of the main trends observed in the data.
- All major functional trends discussed in the paper are now not only tested at the level of the cluster means, but also at the level of individual cells. In all cases, the general trends remain stable. This is systematically documented with statistical details added to the relevant figure legends, and in several cases also with new Figure sub-panels that showing the full data clouds.
- We have also extended the online plotter to allow easy inspection of each individual cell allocated to a cluster in detail.

In addition, we have carefully gone through the MS and addressed / followed all additional comments, in particular in view of improving the clarity of our approach, presentation and interpretation.

We hope that with these extensive changes, the manuscript will be more suitable for publication.

Reviewer #1

The study “Birds multiplex spectral and temporal visual information via retinal On- and Off-channels” investigates how retinal ganglion cells (RGC) in birds (poultry chicks) represent visual information. The study aims at answering whether principles discovered in the mammalian retinal circuits also apply to the avian retina. To address this very interesting question, the authors established an ex-vivo approach for recording light-induced neuronal activity from retinal ganglion cells in poultry chicks using a high-density multielectrode array. To characterize the response properties of the retinal neurons the authors used a rich and established stimulus set which allowed the authors to identify functional RGC groups within their dataset.

The novel and the key finding of the study is that RGCs in chicks represent visual information in a different way compared to mammalian RGCs. While information about the polarity and kinetics of visual signals are represented in a decorrelated manner in mammalian RGCs, this study now reports that these properties are represented in a correlated manner in chicks. This finding is surprising and exciting because it challenges the long-standing view that visual information is efficiently represented by decorrelated visual channels. Moreover, the results question the assumed common architecture of all vertebrate retinal circuits. The presented results are thus of clear relevance for the discussion in the field and will have a significant impact on it.

I very much enjoyed reading the manuscript and the presented data look solid and of high quality. However, I do have several major comments in regard to the presented results and the conclusion that the authors draw from them.

Many thanks for taking the time to review our work, and for the very useful comments!

1) The authors report that most RGCs encode both ON and OFF signals, with ON signals being more sustained and OFF signals being more transient. This is in contrast to what has been reported in mammals and thus a key result of this study.

This conclusion is based on extracellularly recorded neuronal activity and the authors used a spike-sorting approach (Herding Spike2) to assign action potentials to single clusters/neurons. While spike-sorting is a standard and integral part of extracellular recording methods, it works, unfortunately, never perfectly and can and will lead to some level of mixing of multiple single neurons into single clusters. Thus, potentially a OnOff RGC neuron with slow ON and fast OFF kinetics could be the result of combining the spikes from an ON-slow RGC and an OFF-fast RGCs into a single cluster. Therefore, a key conclusion of the study could potentially be affected by imperfect data analysis that results in a mixing of neuronal signals. I do not think that this is necessarily the case, however, in the current form, the manuscript contains very little information on the spike-sorting results that would allow the reader to assess the spike-sorting quality and to convincingly show that there is no systematic bias/error introduced by the spike-sorting.

Therefore, please provide more information on the spike-sorting quality for all clusters, i.e. unit quality metrics and the inclusion criteria etc. Moreover, please also test whether neurons from different functional groups show some sort of systematic bias in regard to the spike-sorting quality. For example, it would be informative to analyze whether the OnOff polarity index is correlated with the spike amplitude or similar quality metric, i.e. whether the OnOff neurons are predominantly found in clusters with small spike amplitudes/clusters with lower spike-sorting quality. Again, I am not saying that this should be the case, but it is important to rule out any potential contamination of the results by the spike-sorting part of the analysis.

We thank the reviewer for bringing up these points.

In the previously submitted MS, our main approach to evaluate the spike sorting quality (beyond metric provided by the sorter itself) was to check how systematic a neuron's response was to a given stimulus. For example, we reasoned that it is unlikely for a poorly sorted neuron to produce a large-amplitude kernel when tested with full field white noise stimulation. Our second approach was to check the electrical image of each neuron and establish if we can identify neurons that have multiple axons which would indicate a mix up of neurons. Our data does not suggest that this is systematically the case.

In response to the reviewers' comments, for all neurons that passed the above tests, we now also calculate new metrics to assess the quality of spike sorting.

These include a "circularity index", a "waveform similarity index" and additional "On-Off quality metrics", as discussed in detail below. Together, these indices suggest that the vast majority of cells were spike-sorted to the best of current possibilities. Likely spike sorting errors were identified in <<1% of cases, and these did not systematically align with any of the presented functional clusters.

Accordingly, it seems unlikely that our results are driven by spike-sorting errors in a major or systematic way.

We have now introduced new dedicated Methods sections where we discuss these spike-sorting metrics, accompanied by a new Supplementary Figure 2 (below).

New Supplemental Figure S2 – related to Figure 1. *a-c*, Examples of three spike-sorted units and their sorting-quality metrics, illustrating a well-sorted unit (*a*), a mediocre unit (*b*), and the worst unit in the dataset (*c*). Shown for each unit are the location of all detected “On and Off-spikes” in space (top left; see Methods for details), a histogram summarising waveform similarities of On- and Off-spikes based on principal component analysis (top middle), and all On- and Off-associated spike waveforms superimposed as indicated (top, right), and corresponding metrics for spikes artificially divided based on waveform similarities as shown (subunits 1 and 0 as indicated). Note how for the cell shown in (*a*), neither the artificial On- versus Off- division (top row) nor the artificial division based on waveform shape (bottom row) lead to different distributions in the remaining parameters. By contrast, units shown in (*b*) and (*c*) show some bimodality in PC space (indicating that two units might have been merged during spike sorting). Note also, however, how none of these examples shows bimodality in On- versus Off-behaviour (which would indicate that an On cell had been merged with an Off cell to produce an OnOff cell). *d-f*, 2D histograms relating spike-sorting metrics (*y* axes) with functional indices used throughout the study (*x*-axis). Note that no 2D histogram shows an obvious diagonal, which would indicate a strong correlation between spike-sorting quality and inferred neuronal function.

Spike sorting metrics In detail:

Circularity. Our new “circularity index” measures the spatial distribution (on the MEA) of spikes allocated to a single unit. Our reasoning was that a “perfectly sorted” cell should yield spike locations that vary with equal variance in all directions (i.e. forming a perfectly circular cloud). To quantify this circularity, we calculate each neuron’s first two principal components of spike locations. If the variance is the same in all directions, variance explained by PCs 1 and 2 should match, i.e. 1:1, while a spatially skewed cell should have more variance explained by PC1 compared to 2. Correspondingly, the circularity index is defined as variance explained of PC1 divided by variance explained of PC2. An index of 1 then represents a perfectly circular spike-location cloud, and larger numbers indicating increasing degrees of spatial skew.

Computing circularity index

PC1, Explained variance: 19.65

PC2, Explained variance: 18.48

Circularity index = 1.06

Based on this index, we find that most neurons in our dataset are well located in space, with a median circularity of 1.6 and a standard deviation of 1.1. 75% of neurons have a circularity index below 2.09, and nearly the entire remainder corresponds to neurons located near edge of the MEA array, where circularity is hampered by the inevitable spatial sampling bias. New Supplemental Figures S2a-c show example neurons of varying quality.

Waveform similarity. We next judged spike sorting quality by assessing the similarity of all spike waveforms allocated to a single unit. For this, we computed the first principal component across all waveforms, and then tested the resultant loadings onto PC1 for unimodality. Specifically, we used Hartigan's Dip Test (Hartigan and Hartigan, 1985) to test for multimodality, such that a high p-value indicates unimodality. The median p value for cells in the dataset was 0.88 with a standard deviation of 0.26. 75% of the cells in the dataset had a p value above 0.66.

Combining circularity and waveform similarity, very few cells (fewer than 1%) failed both criteria, indicating that most cells in the dataset were well sorted. This is perhaps expected based on the quite conservative quality metrics initially applied, that reduced typically 3-6k spike sorted units per recording to around 300. While it is plausible that the very small number of units that failed

both criteria are the results of spike-sorting errors, we are confident that their inclusion did not qualitatively affect the results of our study. For example, the cells were not systematically associated with any specific cluster.

On vs. Off quality metrics. Beyond the above indices, we also directly tested for a possible mix up of On and Off cells into OnOff cells. To this end, for each neuron, we divided all spikes driven by the colour and contrast steps into “On-spikes” and “Off spikes” and analysed each separately. We did not use spikes from the remaining stimuli, because in these cases deciding which spike counts and On and Off was not possible. We then split neurons with a waveform similarity index below 0.9 into two “sub-neurons” according to the shapes of their spike waveforms. We used Gaussian mixed modelling clustering (`sklearn.mixture.Gaussian`, `covariance_type=`“full”) to split the waveforms according to their positions in PC1. Using χ^2 test for independence (`scipy.stats.chi2_contingency`) we checked if the number of On or Off responses were independent from the newly created sub-neurons zero and one (examples in figure x a-c). This was the case for most cells. Of 324 cells with p values smaller than 1, 103 cells had p values smaller than 0.05. Considering that we performed multiple statistical tests, using Bonferroni correction, we found 12 cells with p values smaller than $p=1*10^{-5}$. For those 13 cells (out of >3,987 across all datasets), it is likely that On and Off cells were merged into a single OnOff cell, however in view of their very small number, we think that it is unlikely that their inclusion affected the overall conclusions of the study. As before, these cells also did not specifically associate with any particular clusters.

Other metrics. To rule out additional possible systematic biases in our dataset we also calculated several other quality metrics and moreover checked for other possible correlations in the datasets that could explain a high number of OnOff cells.

For example, we allocated all cells into 10 bins based on their OnOff index, and tested if these bins were systematically related to variations in the circularity index, waveform similarity index or mean amplitude of the spike signal. Since the OnOff index was not normally distributed we used Spearman’s rank (`scipy.stats.spearmanr`) correlation. This revealed a weak but significant correlation between the circularity index and the OnOff index ($c=0.10$, $p=9*10^{-9}$). While this could mean that spatially close On and Off cells had a weak tendency to be mistaken for a single cell because of their strong spatial overlap, it could also mean that on average, OnOff cells had slightly better located spike signals compared to On and Off cells in isolation. In support of the latter interpretation, the weak negative correlation between the waveform similarity index and the OnOff index ($c=-0.07$, $p=6.45*10^{-5}$) indicates that OnOff cells were located slightly better in PC space compared to On and Off cells in isolation. Additionally, there was a weak correlation between the spike amplitude and the OnOff index ($c=-0.057$, $p=6*10^{-4}$). Notably, while these correlations were significant because of the large number of cells, they are all very weak indeed (between 0.1 and -0.07).

We also calculated 2D histograms to survey for other possible correlations between spike sorting metrics and the indices presented in this study (OnOff index, best chirp frequency, TrSus-index, kernel opponency index) and found no clear correlations in the datasets other than the abovementioned weak correlations.

In summary, in view of the resolution limits of the MEA, inevitable limitations in spike sorting and the high RGC density of the chicken retina, we cannot entirely rule out that a subset of On and Off cells were systematically grouped into OnOff cells. However, it seems unlikely that this was a strong effect across the dataset, or that it affected the key results in a major way.

It is also worth considering that our recordings from avian retina are the first of its kind by many criteria, and that this inevitably comes with possible difficulties in data curation and interpretation that will only become fully apparent in time. For example, current MEA systems and retina-appropriate spike sorting algorithms were overwhelmingly developed and tested for work with mammalian and/or salamander retinas, which are more than an order of magnitude less dense compared to chicken. Correspondingly, available algorithms are not optimised for work with high-density retinas. Nevertheless, in view of the above-presented statistics, a systematic mix up of spike sorting errors that could lead to some of the major trends reported in the data seems exceedingly unlikely.

Related, the authors write: “We computed the “electrical image” (EI) for a subset of spike-sorted cells.” and state that this image permits to infer important information about the neuron type, which I agree it does. However, it is unclear how the neuron type was inferred for neurons in which the electrical image was not computed. Therefore, please specify the fraction of neurons for which the electrical image was not available and how the neuron type was inferred in those cases.

Thank you for the opportunity to clarify. In short, our logic was as follows:

- *For a representative subset of experiments (the largest 5), we computed all EIs, and devised a simple algorithm that evaluated whether an axon could be detected in each EI. This indicates that at least 87% of recorded cells have an axon – see Fig. 1k and associated legend.*
- *Since RGCs have axons, but dACs do not, the presence of an axon should positively identify an RGCs. However, the absence of an axon in the EI does not positively identify a dAC. As discussed, this is because the axon could simply have failed to reach threshold in our criterion (e.g. if it ran somewhat distant from the recording electrodes) or be located past the edge of the array.*
- *We conclude that at least 87% of cells were RGCs, but we did not discard the remaining 13% (since they could also include RGCs).*

The fact that we only computed EIs in 5 out of 17 sessions is related to the quite substantial time it takes to compute EIs for a recording. We reasoned that

computing all EIs from the top 5 sessions, which comprised just short of 50% of the cells in the dataset and a good fraction of cells from each functional cluster (cf. Figure 2b), was a reasonable compromise.

We now added the following note to the method's section about electrical footprints...

“Electrical footprints were computed for a subset of 5 recordings containing 1,978 cells (47% of cells in the dataset).”

...and further added a new section to clarify the strategy of filtering the electrical images in the Methods as follows:

“Estimating RGC numbers from electrical images. An axon could be clearly detected in 1,035 (52%) of the 1,978 cells for which electrical images were computed, suggesting that these correspond to axon-bearing retinal ganglion cells (RGCs) rather than axon-less displaced amacrine cells (dACs). However, this RGC fraction is almost certainly an underestimate because many axons were likely missed due to limited signal to noise in a subset of electrical images. Here, signal to noise directly depends on the number of spikes that go into the computation of the electrical image, meaning that the reliability of axon detection should increase as a function of available spikes, as shown in Figure 1k. For example, for 782 of the 1978 cells, we obtained less than 1,000 spikes, which clearly is insufficient to reliably disambiguate the presence or absence of an axon in this dataset. At around 5,000 spikes and above, the number of cells with a detectable axon plateaued at ~87%, suggesting this number as a lower bound for RGCs in our dataset.”

2) The functional clustering of RGC types is another important analysis. While the clustering analysis is described in detail in the method section and briefly mentioned in the results section, important results are not shown in the manuscript which makes assessing the quality of the clustering difficult. For example, the Bayesian information criterion was used to judge the optimal number of clusters but the results are not shown anywhere in the figures. Please provide more information on the clustering and grouping within the manuscript figures or supplementary figures.

Thank you for this comment. In our view, there is no “correct” way to cluster. Rather, we see it as an attempt at identifying useful structure in the data. However, variations of our approach, or a different approach altogether, would almost certainly identify slightly different divisions between the data. Here, the one key thing to aim for is robustness of the big picture that emerges, and we believe we have met this goal. For example, we also tried an entirely distinct clustering approach and got essentially the same main clusters. This clustering approach uses compression-based dissimilarities (CBD) to determine the distances between the responses of individual cells as described previously (Keogh et al., 2007). This approach returned 29 clusters of which many have similar response profiles compared to the ones presented in our paper. To illustrate this match, we determined which of the new clusters contained the most cells from a single cluster presented in the paper and plotted these “neighbouring clusters” next to each other:

Because this chicken RGC dataset essentially presents uncharted territory, we did not go straight to an “optimal” number of clusters as often picked by a BIC, but rather “over clustered” on purpose as a first step – for this we set the algorithm to a fixed number of 100 clusters. This was done to ensure that each initial cluster was reasonably homogeneous, and, importantly, to reduce the chances of missing possible “rare” response types. Here, the BIC was therefore not used to inform the number of clusters, but rather to decide which covariance matrix to use (unshared diagonal, in this case).

As a second step, we then grouped these clusters down to 36 “compound clusters” based on their manually judged similarity. Finally, for simplicity we then removed the smallest 14, each of which composed fewer than 20 cells. In this latter step, we did however ensure that none of the removed clusters carried any notably unique response features – rather, they mainly seemed to represent low signal-to-noise versions of included compound-clusters.

The following figure summarises how clusters were merged to form the final 36 clusters, showing cluster means of chromatic full field flash responses, spectral kernels, and white step responses before (left) and after (right) the merging (we did consider the chirp, but did not plot the responses in this figure, as the stimulus is rather long):

Our logic of going from 100 to 36 to 22 clusters is summarised in the main text as in the below (and further elaborated at depth in the Methods section):

“To identify major functional structures across the dataset we clustered responses to the four stimuli using a Mixture of Gaussian model, followed by manual curation (Methods). This yielded $n = 36$ clusters, of which $n = 22$ had a minimum of 20 members and were retained for further analysis ($n = 3,914$ cells, 98.1%).”

Importantly, we provide full access to the dataset which includes labels for both the original 100 clusters and the reduced 36, and moreover we provide an interactive online plotter for all clusters and cells included in the final 22.

In addition, we have now extended this plotted to allow direct online inspection of each single RGC allocated to a cluster.

3) Comparison to other vertebrate species. Although not statistically significant, the human data appears to follow a similar trend as the fish data (Figure 2d-f). Is the outlier data point in the lower left quadrant in Figure 2f responsible for the statistical insignificance? If that is the case, please further assess the robustness of the results.

This apparent outlier alone does not drive the lack of significance. The data as shown comes out as $p = 0.13$, and with the outlier removed would be 0.067. Moreover, the data as shown are already processed: For the sake of simplicity and comparison between species, we implemented a simple clustering algorithm to bring down the original dataset to 20 clusters, in line with the expected RGC type diversity of the primate retina. Tweaking this clustering step does of course produce different scatter results, however never to a point where the data indicates a significant link between polarity and transience (which would then also require correcting for multiple comparisons).

More generally, this lack of correlation is distinctly expected based on what is currently known about the human retina, and it also comes out quite clearly in the original publication: (Figure 4C in Cowan et al. 2020 Cell):

Please also provide the p-values from the correlations shown in Figure 2d-f such that the reader has access to this important information.

The requested p-values have been added to the legend:

Chicken: $p = 0.000014$ (i.e. <0.001)

Zebrafish: $p = 0.00082$ (i.e. <0.001)

Mouse: $p = 0.41$

Human: $p = 0.13$

Related, why are the values for the transience index below 0 for many neurons in the chick but almost all above 0 for fish, humans and mice?

This is a result of the different overall kinetics in the four datasets, which probably reflect a combination of biological variation and differences in how these datasets were acquired. Accordingly, when designing an index that usefully summarises the kinetic behaviours across all datasets, we had to make some compromises in terms of what exactly “transient” and “sustained” ends up meaning in each retina.

Specifically, the TrSus index is defined as $(A-B)/(A+B)$, where A and B are measures of response amplitudes at two different time points (A being “earlier” than B). Accordingly, the exact timepoints defined by A and B will impact the value of the index (but not the distribution of this index within or across species). As it so happens, some cells in chicken (and also some in mouse!) are very slow indeed – specifically, this is the “slow On” populations. These can take nearly 500 ms to reach a peak. At this point in time, most other cells have already returned to baseline. Because these extra slow cells are the exception rather than the rule when considering all cells and datasets together, we decided to place both A and B timepoints substantially before 500 ms into the response, so as to achieve better resolution of the index for shorter time-periods. However, this also meant that the slow On cells in chicken (and a few cells in mouse) end up giving a TrSus index below 0 (meaning: “facilitating” / response growing over time), even though over the full 2 second stimulation period they clearly reach a peak and then adapt again.

An alternative approach might have been to specifically define the timepoints that A and B capture for each species independently. However, we decided against such a solution as it would make the index more difficult to compare across species.

A condensed version of this explanation is now added to the relevant methods section:

“For each step transition (i.e. On and Off), we also computed ‘transient’ and ‘sustained’ response measured based on the peak spike rate within time windows of 80-160 and 240 – 2,000 ms following the step transition, respectively. These time windows were chosen as a compromise to reasonably capture response properties across all analysed species’ datasets despite their

somewhat different overall kinetics. The exact choice of these two time-windows were judged to not qualitatively affect the results.”

4) The data appears to be from the dorsal retina and not from both the dorsal and ventral retina regions. Given that several studies have recently reported differences between dorsal and ventral retinal in mammals I was wondering whether the presented results in the chick retina generalize across the dorsal and ventral regions. Fully answering this question is of course beyond the scope of this study, but it would still be informative to discuss the presented data in this context.

We fully agree that the collected dataset represents only one very specific state and region of this avian retina, and in time it will be key to understand if and how the properties of RGCs vary in different parts of the eye (or age / sex / time of day / brightness regime etc.). However, as the reviewer notes, we had to start somewhere, and for consistency we focussed only on the dorsal retina for now. We reasoned that this part of the eye (if any) is most likely to reflect “general” properties, because it is usually targeted below the world’s horizon, which in natural scenes is characterised by relatively evenly spread contrast and spectral distribution (unlike e.g. the sky, which can have dramatic brightness transitions that might call for a more specialised type of encoding overall – e.g. see Baden et al. 2013 Neuron). Moreover, the dorsal retina has an approximately average density of RGCs, relative to the whole retina, and it is relatively far from the area centralis and optic disc, where the retina becomes thick which makes it more difficult to get good extracellular signals.

We have now explained this at the end of the step-by-step dissection protocol in the Methods:

“Throughout, we focussed on the dorsal retina due to its approximately representative RGCs density for the whole retina (Seifert et al., 2020), and because it remains far from the pecten, area centralis and optic disc, where the large number of axon bundles and thick inner limiting membrane made the recording of reliable signals increasingly difficult“.

Minor comments:

- Figure 1n: Why are the responses of only 5 out of the 6 wavelengths shown?

The chicken retina was generally poorly responsive at this shortest of tested wavelengths. For simplicity, and to save space, we therefore decided to not include the UV-step response in the main plots (they are included in the online plotter). This is now explicitly noted in the legend to (new) Figure 2, where the responses are first shown.

- Figure 1o: Why is the chirp stimulus strange at high-frequencies? Is this just a rendering issue with the figure or was the stimulus like that?

This is a rendering issue – the figures were unfortunately down-sampled during embedding in the pdf. We now provide the full-resolution figures where the issue does not arise.

- Extended Data 1e: please provide the real number of cells in addition to the circles. While the different-sized circles are a nice visualization, proper numbers would help to better understand how many cells were included in each group.

Done.

- Figure 3d: The authors write that the flicker responses for C1,2 were ON type at low and OFF type at high frequencies. However, Figure 3d suggests that, at least at low frequencies, the responses happen during the downward phase of the chirp, which should be the OFF part of the response. Should the responses not be at the positive part of the chirp at low frequencies if?

These are actually On-responses – the average delay of RGCs in our dataset is quite long at around 100 ms, meaning that On-responses appear to peak during the beginning of the Off phase (for slow oscillations). This effect is complicated by the fact that on average, On responses are sublinear with contrast while Off responses are more linear – meaning that Off responses have tendency to be triggered earlier in the chirp-waveform compared to On (making On- and Off responses part-overlap at some frequencies)

The best way perhaps to explore this is the online plotter:

<http://chicken-data.retinal-functomics.net/>

For example, switching back between Off-dominated cluster 1 and On-dominated cluster 5 shows how both clusters mostly time-align their responses during the very slow flicker, but are anti aligned during the fast flicker.

Reviewer #2:

This manuscript reports and provides interpretation of a large dataset of retinal ganglion cell recordings in the retina of chicken chicks. These are absolutely novel data that add considerably to our current knowledge on the retinal computations that take place in the retina of birds. New findings include – among others - the dominance of on-off channels in the bird retina, as compared to the separation of separate on and off channels in the mammal retina, and a suggestion how both chromatic and achromatic information may be coded in the bird retina. Already the large data set reported here is worth publication, but the authors go beyond this and compare their results with those from other vertebrates. While some of the interpretation is maybe speculative, such parts are clearly marked, and the conclusions are well founded in the results. Methods are complex but mostly well described, and the development of a methods allowing reliable recordings from bird retina is a major milestone also for future work on bird retinal coding.

Clearly this paper should be published, as it adds important new findings on a long under-studied system. My two minor comments concern (1) missing details on methods in the main part of the manuscript (which make it more difficult to understand for a general reader), and (2) the perspective that seems to largely take a mammal-centric perspective, while evolutionarily, birds are likely more similar to an ancestral state of vertebrate retina, as they have not lost two cone types. This evolutionary interpretation is given in the end, but it would be more convincing to start with it, and treat the mammal situation as derived from an ancestral state rather than “the state of knowledge” from the start.

Else, the paper, despite long, reads well, while the figures, specifically the first figure are crowded making it sometimes difficult to really appreciate the data. If possible, making two figures from figure 1 (with larger part figures) would probably improve the paper.

All of these points are minor.

Thank you very much, in particular for helping us improve the clarity of our MS.

Detailed comments

The abstract: it is written as if for a specialist journal – it may be good to start with a more general sentence clarifying that the paper is on vision. For instance, there are animals other than vertebrates that have visual systems, but the first sentence is written as if it was general, and then only in relation to other vertebrates. The word “bird” appears in the second- last sentence only. Keep the general reader in mind for the abstract

Thank you, we have reassessed and tweaked the abstract to reflect these comments. For example, we have clarified in the first sentence that the paper is about “vertebrate vision”. However, we think that it should be clear that the paper is about birds, as this is the first word in the title.

We also think that, on balance, the overall “mammal centric” narrative is suitable given that most vertebrate vision researchers (for better or worse) work on mammals (see also our more elaborated response to a similar comment further below), and certainly by far the most comprehensive understanding of “how vertebrate retinas work” comes from work on mammals, especially with regards to neurophysiological studies of retinal circuits. Accordingly, we think that a mammalian perspective presents a useful starting point when trying to understand how other vertebrates’ eyes and brains might work.

Line 13: an English sentence cannot start with “But”, please reword

Done.

Highlights: I am not convinced by the first one. What is “visual function in an avian retina”? Could it be more telling and specific, maybe First large scale survey of retinal ganglion cell function in an avian retina. One could argue that cone sensitivity is also a “visual function in the avian retina”, and this has been studied for a long time.

Changed as suggested.

Introduction:

Like the abstract, this is written for specialists, it may be difficult for a general reader to understand*.

*For instance, line 90: “chicks represent information about the polarity, kinetics, and wavelength”. First, say chicken (they also have a Latin name which needs to be mentioned at first mention) chick means just a young bird, you studied chicken chicks. You need to first say the full term, then you can continue with just “chicks”.

Done.

Second, it is very colloquial to say that chicks represent... - the signals recorded from cells in the RGL represent the properties of visual stimuli. Visual stimuli or light stimuli needs to be mentioned in the sentence, else it makes no sense. Sorry, colloquial language can make reading difficult.

Changed as suggested.

At this point, the reader has not yet been told what on- and off-circuits are.

We think that the terms of On- and Off- are sufficiently well-established that a reference should suffice in this instance. Accordingly, at the first mention of On and Off circuits we cite Westheimer’s’ classical review on what On- and Off-circuits mean and how they come about.

Line 73: while the zebrafish work from the Baden lab is amazing and admirable, they

were not the first to study fish retinal circuits, and it would be fair to mention some older work, at least in passing.

References updated.

Line 76: “retinal functions” is a very broad term, which – in my understanding - includes everything from oxygen supply to visual pigment turnover, may be better to say what you mean: the coding of information by the inner retinal layers.

Changed to “retinal computations”.

It may be helpful to mention for a general reader which cells a retina has and which fulfill the functions mentioned above and below.

Since we do not record from (or discuss at any length) the horizontal cells, bipolar cells, or “regular” amacrine cells, we think that the existing reference to general reviews on retinal organisation should be sufficient in this case. The identity and roles of the remaining classes (photoreceptors, displaced amacrine cells and ganglion cells) are now briefly explained as part of the retinal schematics shown in Figure 1.

Another important piece of missing information – at least for a general reader but maybe even for primate vision researchers, is the evolutionary aspect. This entire data set and its comparison to mice, primates and fish, is most interesting from an evolutionary perspective. As you are aware, while fish may be closest to the ancestral visual system (even though bony fish may have moved quite a bit from that situation), birds are more similar to them as they have kept the full complement of visual pigments and thus, receptor types (even though, it may be debated that double cones are new, compared to fish), while mammals have lost two cone types and likely, with them, retinal neuron types. So chickens are much more general than any mammal, and primates even have a visual system that uses a spatial coding channel to code colour, which makes them the least general of all, and all difference to them may be quite expected. Salamanders and turtles are likely similar, but the best reason to study birds and not these in detail, is that the birds are the absolute vision-specialists among vertebrates. I think there needs a short mention here to prepare the reader. Otherwise, the reader may ask, why do we need to know anything about chicken in the first place?

We agree that the evolutionary aspect is an important aspect of this work, and correspondingly this is included at various points throughout the MS (for example at the end of the abstract and in the last figure and much of the discussion). However, as argued above, we believe that it does make sense to start out from a mammalian perspective, simply because this is where we currently have the best understanding of how (these) vertebrate retinas are functionally organised.

Results

Line 130: you use “n” for both the number of preparations and retinae – would it be better to use “n” and “N”?

Done.

It will be helpful to mention the size of the retinal pieces as the methods section is far away.

Added.

Line 134: displaced amacrine cells: first mention here. I really think the retinal cells that will be relevant for understanding the results need to be mentioned in the introduction.

See above. In addition, we have here added that dACs are “local interneurons” which contrasts them from ganglion cells here described as “the retina’s output neurons”

Line 136: just a side note: I really like the presentation of these ‘electrical images’, very helpful! In the figure, is it possible to indicate some information on the orientation of the cells in the retina (what is up and down, right and left, as you mention in line 140 that the axons of RGCs should be oriented towards the optic disk – how would you know?).

We deleted “projecting to the optic disc” because strictly speaking, we do not know that (it would however be quite surprising if this was not the case).

Unfortunately, we did not always manage to maintain a known orientation of the retina in the chamber (this can be hard to do if the retina rotates at the final mounting step where we would have cut away all landmarks that one might otherwise be able to keep). In a way, the logic might therefore be the other way round – i.e. using the orientation of the “computed” axons (which do always point in the same general direction in our experiments) as the landmark to later re-orient the retina. However, since we did not present spatially patterned stimuli, the orientation of the tissue in the chamber does not affect the results, and therefore it seemed best here to remove the statement about orientation.

Would it make sense to also show an example of a displaced amacrine cell, maybe in the supplement?

We did consider this; however, one problem was that we cannot be sure that a cell that lacks an obvious axon on the array is in fact a dAC – it could also be an RGC whose axon we did not detect (e.g. because it was insufficiently near the electrodes). For simplicity we therefore opted to not include dAC examples in the figures.

Line 155ff: this is general information on the chicken retina, which I would have placed in the introduction. I find it strange to place it here.

Thanks, but we think this is the best place, because it is here where we justify the placement of the LEDs, which only make sense in the light of what we know about cones. We jointly summarise the cones, their spectra, and that of the LEDs in Figure 1.

Lines 167ff: even though stimulus steps are given in the methods, can they be repeated here (n photons or photons per s or something like that, in parentheses)? More than that, what does “corresponding” mean for the colour steps – same number of photons, or same number in the wavelength range? Important to know here without having to search in the Methods.

Added as suggested.

line 204: quite frustrating not to get any idea of what “Mixture of Gaussian Model” means unless I move to the Methods part. Is there a short way to say it here?

Added as suggested

Line 225: “This revealed surprising trends” is a strange way to start a description, as the reader is not aware of the authors’ expectation and thus, would not know what should surprise them. Is it possible to first describe the results and then why they are surprising? Moreover, in which sense is the word “trend” used here? Often, it is used for “just not significant” results, something I dislike, so please clarify how the word is used here.

We rephrased this part, removing the statement regarding “surprise”:

“This revealed that cells in most clusters responded both to light onset and offset, rather than forming well-segregated On- and Off-channels.”

Line 226: I guess cells responded to ON and OFF. Clusters have earlier been described as responses, and responses do not respond. And certainly, clusters don’t respond. Cells grouped in clusters respond. Thanks for clarifying the writing.

Rephrased as suggested – see also our response to the previous point.

What is a “large-amplitude kernel, in terms of cell responses? Does it mean that cells responded with high spike frequencies, or to high amplitudes of stimulus intensity?

Large amplitude in this context means that the average stimulus that preceded a spike in a cell/cluster deviated strongly from the mean (ie. a flat line). Accordingly, the kernel represents the average stimulus that preceded a spike, and there are, in principle at least, many ways to generate a “large” or a “small” kernel. For example, a “noisy” cell will usually give a small kernel because stimulus-unrelated spikes (noise) will average out to a flat line. However, a small kernel could also result from a cell where spikes can be triggered by

“opposite” stimuli (e.g. a perfectly balanced OnOff cell, by definition, would also give a flat kernel). Accordingly, as also noted by reviewer 3, these kernels do not necessarily translate into more intuitive metrics such as gain or sensitivity. It is in part for this reason that we presented a variety of stimuli that together reasonably describe a cell’s response. For example, the steps highlight On-Off responses in a way that the kernels cannot, but at the same time the steps are high contrast from black, while the kernels probe responses around an average grey background.

I generally think it is more correct to say “high amplitude”, not “large amplitude”.

Changed as suggested.

It is indeed a bit confusing what we mean with clusters. In line 210 we say that “To what extent the 22 clusters correspond to anatomically and molecularly distinct types of RGCs and/or dACs is unknown.” I guess we should add before that that we assume that the clustered cells might correlate with specific types of RGCs.

We think of the clusters as a simplified description of the data, which allows us to boil down the big trends into a manageable numbers of response archetypes.

In an ideal world, each cluster might correspond to a bona fide type of retinal ganglion cell, however we know from work on other species and other neuron populations that this is almost impossible to achieve except in some of the most clearcut cases (e.g. it works, to some degree, to pick out some of the major RGC types in primate, but even here it fails after the first 5 or so most clearcut types are found). In contrast, our chicken dataset is quite complex, and possibly noisy, and we also have no usable a priori information on the types of responses and cells that we might expect, to e.g. guide the clustering strategy. Consequently, all we can meaningfully hope for from the clustering approach is to identify some of the more major structures in the data.

For these reasons, we hesitate to set the problem up based on the expectation that clusters and RGC types will strongly correlate.

Line 287ff: the comparison is rather superficial. Neither are potential reasons mentioned nor how the groups are similar or different. I would add both either here or later. A major reason for differences is the fact that mammals lost two visual pigments and thus, two types of cone, and thus, may have lost horizontal, amacrine and ganglion cell types.

In this part of the results section, we think it is appropriate to highlight some of the major differences between RGC light responses across species, without already here speculating as to how they might arise.

We fully agree that the loss of the RH2 and SWS2 cones in mammals are probably a central contributing factor to some of these differences – however this remains a correlative observation that needs to be followed up (for example by interfering with those cones in species that retain them and observe if and

how that changes their retina's functional organisation). We do however offer this possible explanation as part of the discussion (section "Why are chick and mammal retinas so different").

Line 295: I have no clue what – quantitatively – a “sizeable complement” is meant to say. Can you please give the reader less cloudy wording, and, preferentially, also numbers?

We have added numbers as requested: (~65-70% and 30-35% in chicks and zebrafish, respectively).

Lines 414ff: I think the two indices are a very useful way to describe the data. Therefore, I am wondering whether these have been newly invented for this paper, or are used elsewhere – no references are given, so I guess they are new. If that is correct, it would be interesting to know whether there is any way to compare the results to those obtained from other vertebrates (as was done with other aspects in earlier sections).

Thanks, yes, the indices are “new”.

It might be possible to compute something analogous for zebrafish (based on our lab's unpublished data), however complementary data from mammals would be difficult to find (and certainly would have to come from different studies than the ones used for previous comparisons). To our knowledge there is no study on mammals that recorded from a large (representative) fraction of an animals' RGCs and probed all those cells at a finely resolved spectral scale. Since as the reviewer notes mammals only retain two on the four ancestral cones, most studies interested in mammalian colour vision tend to correspondingly pick two wavelengths at which the cones' opsins show limited overlap. This makes for a powerful strategy to probe each cone system in isolation but does not provide the necessary spectral resolution to compute these indices.

Line 546: “For of the Off-channel, this trend extended to blue” – remove the “of” here.

Done.

Line 553ff: I am not sure about the comparisons between the different colour channels; what do they say given that unfiltered red cones have broad sensitivities, and that the relative intensities of the four coloured stimuli are unlikely to be matched for any of the receptor types. They may be descriptive of the data here but may not tell much about the chicken retina, so it may be good to be more cautious here? For instance, a response to white light may be more similar to a response to red light, as the majority of cones in the retina (double cones plus red cones) are red-sensitive. The sensitivity curves of unfiltered cones overlap considerably, making differentiation less clear than it will be in the real retina with oil droplet filtering. These points may have to be pointed out when interpreting results.

We agree in principle, however we think that any spectral differences that can be observed already in this “spectrally unfiltered” state should if anything become exacerbated if filtering were to be added. For example, the finding that red responses are larger/quicker than white responses implies that on average, circuits served by red-cones (single and/or double) are inhibited by circuits served by the other cones. If these circuits were now activated with less overlap (because of added spectral filtering), the basic expectation would be that the response differences between “white” and “red” should increase.

Clearly, there is much follow up work to be done, especially on the ideas surrounding the encoding of spectral information using temporal information (classical ideas tend to revolve around time-independent opponency), however for now, as the reviewer notes, all we can realistically do is describe the data. This is also why we approach the same problem using three different sets of analyses that also include two different parts of the data (kernels versus steps).

Line 554: The “red response” is later (line 569ff) used as reference, so it may be important to clearly define what it is – the response to the highest step of red light? Please define here to help the reader understand exactly what was done. The other responses, I assume, are defined accordingly, which also needs to be said.

For the colour steps (unlike for white), we only presented 100% contrast. This is why we compare the colour step responses to the 100% white step here (as opposed to the other white contrast levels). This is now clarified when first introducing the stimuli.

Line 587: unclear why, after “all....” , the word “step” is singular, please check the sentence, moreover, in the next line, please add “white” if you refer to white-light steps there, else please specify.

Thanks, done.

Line 584: not sure I have seen a definition of “temporal” envelope. While it is likely clear to many readers, it may not be clear to all,

Thanks, added: “(i.e. the evolution of step-responses over time).”

Line 594: while clearly, the two types of analysis (amplitudes and latencies versus PCA) are different, the PCA also uses amplitude and latency information, and thus, results are not independent. As is written, it is not clear whether a PCA of just amplitude and latency would give the same result, and thus, which contribution the “temporal envelope” makes.

We tried this, and as predicted this it gives similar results. However, comparing the magnitude of these differences in a meaningful way is not trivial since the input space is different. Accordingly, for simplicity, we decided to not include this additional analysis into an already quite analysis-heavy section of the MS.

We have now reworded the introduction paragraph for the PCA to clarify that this analysis is not fully independent of the simpler amplitude/latency analysis, but rather more comprehensive:

“Beyond amplitude and latency, step responses of the OnOff cells differed in their overall temporal envelopes (i.e. the evolution of step-responses over time). Accordingly, to capture amplitude and kinetic differences more comprehensively across ‘coloured’ and ‘white’-step responses we used Principal Component Analysis (PCA).”

Line 617: in a results section, I would not speculate on “perhaps also in some other non-mammalian species” – Maybe you could generalize “and perhaps also in other birds” but anything broader makes little sense here, and seems to reflect a very narrow mammal-centric perspective.

We deleted that part-sentence.

Discussion

The discussion also starts mammal-centric. Not sure this is meaningful unless the authors submit the paper to a journal mostly read by mammal vision scientists.

For better or worse, we think that most journals are mainly read by mammal-centric vision scientists, simply because currently most vision scientists work on mammals. We therefore think that a mammal-centric perspective is appropriate, if only to highlight as clearly as possible how non-mammalian eyes appear to at least in part work by different principles.

For example, from our recent survey of vertebrate retina publications since 1950 (Baden 2020, Sem. Cell Dev Biol.):

Figure 1 | Retina-publications 1950 - 2020

This does not include the much larger communities that work on central visual circuits. For example, by the same search-metrics, there are currently ~10,000 publications per year on mammalian visual cortex alone (more than all retina papers from both vertebrate and invertebrate species combined).

Line 674 f: this is an odd sentence, maybe reword? The entire paragraph is speculative (nothing has been learned about space, in the present study so why speculate?) anyway and could also be removed.

We deleted the paragraph as suggested.

Line 741: I thought frogs also have double cones (e.g. Nilsson 1964 but also Donner and Yovanovich 2020) – are you sure only birds and reptiles have them? – Only birds have double cones that always express LWS in both members, reptiles as well as frogs have different combinations of opsins expressed in both members. And while they are independently wired, there are good data indicating both members are connected by gap junctions, maybe worth mentioning as it must have functional implications.

Unfortunately, this relates to a long-standing dual use of terminology in the field. In essence, there are two different cone-phenomena in vertebrates that have been called “double cones” in various contexts.

One, as the reviewer notes, are the in our view misnamed “double cones” as found in many fish and amphibians – so called, because the ancestral green- and red single cones have a tendency of physically pairing up in some species. For example, in zebrafish adult cone mosaics, red and green single cones are strictly adjacent to each other. However, importantly, these are still, phylogenetically speaking, single cones. This can be readily ascertained by considering development. In larval zebrafish, the same cones do not associate but rather form independent mosaics, with unequal retinal distributions and stoichiometry. They only begin to pair up in juveniles. Moreover, the associated cones not only express different opsins (RH2 and LWS), the rest of the two cells are entirely distinct and independent, for example each serves only part-overlapping postsynaptic circuits that are in line with circuits of single cones as found in other species (for example they make the same cone-type specific horizontal cell connections, no matter if the two cones are paired up or not). This argument is summarised, for example, in Baden 2021 Curr Biol (cited).

By contrast, only birds and reptiles have what we might call “true” double cones – that is, a pair of cones that express the same LWS opsin, that are always found in association, and – critically – that are distinct from the red-single cones (which are also present). Birds therefore have 6 cones: four ancestral single cones, plus double cones (double cones should count as two, since their principal and accessory members are different cells, and they make independent postsynaptic connections). Notably, these also come out as 6 in transcriptomic studies – e.g. Yamagata et al. 2021 eLife - (in contrast to those in zebrafish, which are clearly just the 4 ancestral single cones, e.g. Angueyra et al. 2023 eLife).

Line 767ff: very interesting paragraph, just an additional thought: Maybe, LWS off is a very good reference, as it is a highly sensitive channel?

Absolutely – sensitive in the sense of spectrally broad. In terms of gain, we would probably expect the short-wavelength system to be dominant, since they use light-focussing clear oil droplets, and perhaps more importantly, since short-wavelength photons have higher energy which makes it easier to detect them before running into thermal noise. There are many interesting considerations here that could be added, however in view of space we have decided to limit the discussion on these points.

In addition, this may be a reliable way to make the chromatic channels independent of intensity.

Again, yes (!) - although this would possibly also work with other forms of opponency. Understanding these ideas more fully will likely require research across multiple species and levels of organisation.

Line 785: equally interesting but “cortex-like” is not convincing – reminds me of a very naïve interpretation of stomatopod photoreceptor tuning being cortex-like. It is always good with the most simple explanations, unless they can be disproven, and nowadays, there is a lot of evidence (from invertebrates as from vertebrates) that already at the first synapse in the retina, the spectral signal may be influenced by opponent input from other spectral types, in vertebrates mediated by horizontal cells. If this is what you think happens, it would be good to mention it and give references.

We use the term cortex-like in view of the pioneering studies from the 1980s and onwards on primates where researchers found “narrower than opsin” tuning functions in the cortex that could not be observed at earlier stages of organisation. The same terminology seems to be quite routinely used in the field – e.g. Kinoshita et al. 2022 Curr Biol: “Cortex-like colour-encoding in the mushroom body of a butterfly”.

Beyond terminology, we think that this may in fact be a general principle of spectral coding. While opponency is useful in separating wavelength from intensity, once this computation has been implemented, there is arguably no obvious reason why the same opponency must be mirrored in all subsequent processing layers. It is perhaps more energy efficient, and simpler, to rectify the system, thus clipping the “negative” part of the opponency. The resultant cell will still be approximately white invariant because its input is white invariant, however it will also be narrow (because the positive lobe of an opponent cell is narrower than its upstream positive drive). This is what these “hue cells” perhaps represent, and it appears that in chicken at least, already some RGCs have such properties. Essentially a “bar-code” system for wavelength (if a sufficiently large and diverse population of such spectrally distinct channels can be built).

Notably, this is very different to what we currently understand to be the case in mantis shrimp vision. There is (to our understanding) no evidence that their narrow tuning functions are the result of a preceding opponent computation. Rather, they seem to come about by spectral filtering. Consequently, they are not expected to be white invariant. (To our understanding it has not yet been possible to directly assess these stomatopod photoreceptors’ light responses either.)

Methods

Animals: can you give the age at which chicks were sacrificed, please?

Done.

Light stimulation: it is important to give at least some intensity measure of the used light stimuli. It may be useful to express intensity in terms of the number of photons per time unit and visual angle and area reaching the level at which the retina was positioned.

The LED brightness levels are given in nW (i.e. radiant flux), a fairly commonly used measure in retina research. These numbers, as requested, are now reiterated also in the results section.

This measure can of course be converted to other typically used measurements, such as the one suggested (which is probably the most correct way of doing this, but at the expense of making it quite complicated), or perhaps more typically for vertebrate retinal physiology work, into R^ (i.e., the expected rate of photoisomerization per second and photoreceptor). Depending on the type of photoreceptor, a ~60 nW LED approximately translates to what we might call “low photopic conditions”, so 10^4 - $5 R^*$. However, there are many assumptions in this type of calculation, which specifically in birds, where variables are not consistently known, makes R^* a somewhat less useful value. Accordingly, we think that nW is a reasonable measure, now alongside a statement that this approximately corresponds to low photopic conditions.*

It is also important to know how the intensity of the single LED stimuli related to the intensity of the white light stimulation.

“100% contrast white” in this case simply meant all LEDs were maximally activated concurrently. This is now clarified, as requested, in the results section when first introducing the stimuli, and reiterated in the Methods.

Line 1077ff: while it sounds likely that the selection of methods was sound, saying “we found that unshared diagonal covariance matrices provided the optimal solution” may leave the reader wondering how “optimal” was decided upon. Is it possible to clarify? A methods reference may also help.

This was judged based on the BIC, as now explicitly clarified:

‘The optimum clustering was judged to be that which minimised the Bayesian information criterion (BIC), which balances the explanatory power of the model (loglikelihood) with model complexity (number of parameters). Using the above procedure, we found that unshared diagonal covariance matrices provided the optimal solution (i.e. the solution with the lowest BIC).’

Line 1108ff: similarly, is there any way to explain whether you expect a different result in any way, if you modify the analysis but, for instance, changing time basis (eg only taking the first 1s after stimulus on or off)? Robustness of results is important.

The exact choice of evaluation timepoints (if they were in a reasonable range) did not qualitatively affect results. For example, a cell that reliably spikes during the On-step will come out as “On” no matter if the evaluation time window was 1 or 2 seconds, so long as at least some spikes occurred within the first 1 s (which typically is the case for all On-cells). However, since the steps were 2 seconds in duration, it seemed the most unbiased approach to take the full 2 s response window into consideration. We now emphasise that the 2s window

used is the “full” step duration. For anything more involved, we provide the full dataset for download, so interested colleagues can play with these kinds of indices to explore these kinds of structures in the data.

Figure 1 (and general) vision scientists may be aware that there are colour-blind and colour-deficient readers who need to understand the figures as well, so in addition to colour-coding stimuli, please add a letter somewhere to explain the stimulation wavelength. Please check for all figures. Interestingly, the occurrence of colour-deficiencies seems to be higher in vision scientists than in the general population.

All figures have been checked for this issue and fixed, where needed.

More generally, I find figures very crowded and wonder whether authors are forced to combine so many part figures in days of electronic journals, for which space should be less of a cost. Some part figures – important ones such as 1m-1p) are so small that readers will miss a lot of important and interesting details. If possible, can you give these an own figure and make it a little larger (not all of them in one row)?

It seems that the figure resolution in the provided pdf was quite low, which probably led to some of these difficulties. This is now addressed by providing full resolution figures. However, we also agree that Figure 1 was quite dense, so we have removed the example light responses into a new Figure 2, where we also rebalanced the allocated space per stimulus to better highlight key features of different response types.

Figure 2: d-f: it is said “as b” but the clusters are all grey – so are there only sustained ON responses shown? Else, please add that there is no colour code here. Please also explain the lines in d-f.

There is no colour code here. This has now been clarified in the legend as requested.

Reviewer #3.

Review of NCOMMS-22-49472, 'Birds multiplex spectral and temporal... '

This is an interesting study of population-level visual encoding by ganglion cells in the chick retina. The authors developed an approach to record chick retina on a high-density multi-electrode array (MEA) and studies the responses of all isolated units across the recorded area to a stimulus with broad spectral and temporal content. This approach mimics that of earlier, highly cited work by some of the same author in mouse, and allows reasonable comparison between results between an avian and a mammalian species. The manuscript has a strong comparative component, which is both appropriate, and timely considering the current acceleration of knowledge of mouse retina (mammalian) where, despite initial discoveries and decades of research, amphibian, avian, and fish retina have not entered this somewhat new era of comprehensive functional study (with some exceptions).

The authors reasonably use cluster analysis to group isolated units. The number of units is less than apparent number of ganglion cell types reported from gene expression study, so that clusters may contain multiple types with similar response properties – acknowledged by the authors. The authors report that ON-OFF response types predominate, and broad groups – ON, OFF, and ON-OFF – differ in their temporal response tuning and encoding, as well as in their spectral tuning: OFF response are fast and largely achromatic, ON responses are slower and carry the bulk of the chromatic information. ON-OFF response types carry both signals, and this is reasonably interpreted as multiplexed encoding, where the early, fast OFF component can be decoded separately from the later spectral ON component. Comparison across clades is extensive and well done and the overall volume of new information about chick retina function considerable. I find this a solid study that helps broaden our current fairly (dangerously?) narrow, mouse-centered understanding of retinal functional organization.

Thank you for taking the time to review our work, and for your kind support.

I have the following comments.

1. Multielectrode arrays have a recording bias. Unless I missed it, this, and its possible impact on the results (number of clusters? Over-representation of some types) is not addressed in the manuscript, but some explanation of the possible impact would be appropriate.

We fully agree, and in response we have now emphasised this possibility in the results section where first discussing the observed functional diversity and clustering as follows:

“Similarly, inherent to extracellularly recording the activity of many densely packed neurons in parallel, our MEA dataset likely comprises a sampling bias. Considering the high density of neurons in the GCL of chicks (Seifert et al., 2020), we estimate that on average our MEA and spike sorting approach

(Methods) yielded signals from ~10 – 20% of all RGCs on the array (out of ~30,000), but subsequent filtering by a series of conservative quality criteria (Methods) further reduced this number to about 1% (~300 well-isolated units, cf. Supplemental Figure S1e). Consequently, the reported proportions of functional types may not accurately reflect their proportions in the animal, and some functional signatures might be missed altogether. In the absence of complementary data obtained by different recording techniques (e.g. by 2-photon imaging, or single electrode sampling), it is currently not possible to assess the size or nature of this bias.”

2. Related, MEA ‘shows’ cells in a recorded area. Morphology would, too. What % of the morphologically identified ganglion cells is recorded on the MEA?

Thanks for pointing this out. The somata of RGCs in chicks are quite small and densely packed, and we expect each electrode to be directly adjacent to several individual RGCs (perhaps in the order of 10 or so, plus then there are gaps in between the electrodes where cells will be more likely missed as well). As also noted by reviewer 1, this of course leads to possible issues with spike sorting (e.g. how many units to expect per electrode?). Consequently, we took a very conservative approach in spike sorting, counting only the largest/most obvious units. The consequence is that we miss many cells. We estimate that in a typical experiment, we pick up well-isolated signals from ~1% of RGCs on the array. This is now noted as part of the new section copied above.

(As noted above, we probably pick up less well-isolated signals from perhaps 10-20% of RGCs but including these in deeper analysis would yield a lot of noise and uncertainty. We do of course store the raw datasets, but they are too large to make them routinely available online: in the order of 9 GB per minute of recording so ~0.5-1 TB per 1-2 h recording. We are of course happy to share them with interested colleagues upon request, probably by posting hard drives.)

3. The authors use reverse-correlation analysis to measure visual response kernels for the recorded units. Reverse-correlation measures linear response properties, whereas ON-OFF responses, for example, are obligatory nonlinear. How is this dealt with in the analysis, and what may be missed?

The reviewer is right: The kernels alone give a necessarily limited view of the full response properties of RGCs. To part-circumnavigate this well-recognised issue, we do not only rely on the linear kernels in our analysis, but where possible draw complementary information from the step responses, which do not suffer from this limitation in the same way (they have other issues, of course). For example, in Figure 5 (now 6) we look at the possibility of multiplexing based on each stimulus’ dataset individually to reach essentially the same conclusion.

In general, we chose to present both the very “simple” step stimuli and the more complex noise stimulus because together they usefully probe somewhat different response features – the linear kernels give a first approximation of the “main” thing that drives each cell, while the step responses give a better idea

of nonlinearities such as the On-Off balance. Clustering is then performed based on PCA features taken from both stimuli (in fact the kernels are essentially flat for all On-cells, so using kernels alone would have discarded these cells).

A more comprehensive option for using the data from the tetrachromatic noise stimulus might be to also compute the static nonlinearity for each kernel, or better still, to start with some form of covariance analysis to tease apart if and how different linear (and nonlinear) response aspects interact in each cell. We have in fact done this for a subset of cells, however the results were extremely complex: Depending on parameters used for analysis, responses from single neurons could be computationally broken into sometimes dozens of seemingly independent response components:

Figure above – example results from covariance analysis for a single chicken RGC, showing the first 3 (i.e. most important) components, as well as numbers 79 and 80. Note that each component, even down to the last shown, carries some structure that could be considered as “meaningful”.

Clearly, inclusion of this type of extra analysis would dramatically inflate the complexity of the dataset, and therefore probably deflate its initial utility. Accordingly, we decided in this first overview publication to focus only on the very basic first step of these computational possibilities, i.e. the linear kernels.

We now clarify this in the methods section where detailing how kernels were computed:

“Note that the spectral kernels represent the linear part of each cell’s response, meaning that non-linear effects (for example On-Off behaviour) are not captured. For simplicity, we here focussed only on the linear aspects of noise-responses, and instead draw information about nonlinear properties of stimulus encoding from the responses to the complementary step-stimuli (see above).”

4. A major issue with the results as presented is that most analysis is performed on cluster averages, i.e., the average response waveform of all units assigned to a cluster (line 218). Because of this simplification, variance on the responses within each cluster is gone, which obviates statistical comparison of response features across clusters. This is a major problem that impacts many claims, beginning with the results of Figure 2, and must be addressed. The results are (cleverly?) introduced as ‘surprising trends’ (line 225) but are presented as more than ‘just’ trends throughout. Please address.

Thank you, we agree – collapsing the data into cluster means does sacrifice immediate access to information about the variance across cells within each cluster. As further elaborated also in response to the other reviewers, we see the clustering and their results as a tool to identify structure in the data that allows us to explore and present the results in an understandable way.

However, importantly, all trends presented and discussed throughout the MS that are perhaps most readily illustrated based on the cluster means do hold also when considering the full data-clouds of individual responses. This is showcased, for example, in Figure 5/S5 (previously 4/S4), where we show the cluster means for our different “colour indices” in the main figure, but the corresponding full data clouds in the supplement.

Since the clusters probably only approximate the “true” RGC type diversity anyway, we think that appropriate statistics, if not using the cluster means, should be done on all cells, rather than on the variance within each cluster (where the variance is mainly related to the clustering than the dataset as a whole). For this reason, for example, we include the full data populations’ error shading in Figure 2b, which shows that the trend linking polarity and transience is quite robust also when looking at the raw data. In a previous version of the MS we had included a plot of the full data cloud:

*With a correlation test, this data comes out with a correlation coefficient of -0.35 at statistical significance $p < 0.001$ (in fact, $2.7 * 10^{-117}$). Accordingly, both when*

using the cluster means, and when using the individual data points, transience and polarity are anticorrelated in the chicken data. We now provide this plot (new Fig. S3b), and give additional statistical measures for each key trend based on the full dataset, independent of clusters, throughout the results section. We also added a new set of plots to illustrate how frequency coding scales with polarity – this makes a similar point to the above, however in this case is based on the chirp data which probes frequency coding more explicitly compared to the step:

This is now included in few Figures S4c,d.

Together, we hope this captures a more comprehensive demonstration of the key trends reported throughout the MS.

Beside these additions, we do show error shadings in clusters' example plots (e.g. Figure 4g, j-m, Figure 5a-l, l,m,), alongside histograms that use the full data distribution (e.g. Figure 4i), except in cases where the data density is so high that it would make reading the plots difficult (e.g. Fig. 4d) or when using the cluster means as the starting point for additional statistical analysis (e.g. Figure 6), In this latter case we think that including the variance of each cluster would make this already complicated analysis quite difficult to read – however the core idea certainly works also on individual cells - perhaps best appreciated when considering the kinetically very different step responses and kernels of individual neurons as shown in Figure 2). Moreover, and probably most importantly, we showcase an interactive cluster-wise online presentation of each cell which should give a very good idea of the variance within each cluster to the interested reader, and which can also be downloaded for further scrutiny.

This has now been further extended to allows online selection and viewing of single cells https://chicken-dataset-plot.herokuapp.com/apps/single_cell_plot.

5. Line 63: a study by Ratliff et al., PNAS 2010 (with additional relevant references in Introduction) shows that the division of visual encoding in the retina is distinctly not equal. Consider qualifying this statement.

We agree that there are many aspects in visual encoding, also in mammals, which are not strictly balanced. However, as argued below, we think that they tend to be substantially “more balanced” than what we observe in chicken.

Regarding the mentioned paper (which argues that natural scenes and retinas both are “dark-biased”): we agree that under some circumstances, some aspects of natural scenes can be dark-contrast biased. An intuitive example of this might be an image of a clear sky with a superimposed silhouette of a bird. To a weaker extent, a dark bias also exists below the horizon in some terrestrial scenes, especially if sufficiently zoomed in onto vegetation reflecting the sun, as used as an illustrative example in this study. However, we think that the full picture is substantially more complex and varied:

For example, depending on the spatial scale and waveband, many “cluttered” natural scenes are in fact approximately balanced in terms of On- and Off-contrasts. We showed this some time ago on the example of below-horizon mouse-view scenes (Baden et al. 2013, Neuron), and it is also the case for many other natural scenes we have since looked at (unpublished). Other scenes still are distinctly light biased – for example when looking up into a dense canopy with the sun peeking through, when looking along the horizon of a forest that is dominated by tree-trunks rather than shrubbery, or depending on the angle of the sun, when sitting in high grass or a bush. Clearly, whether natural scenes are light- or dark biased, or balanced, is strongly dependent on the specific natural scene in question.

Something similar could be argued for mammalian retinas, which, if anything, are weakly On-biased (not Off, as claimed in the paper): For example, if we count up the type-diversity of retinal cone-bipolar cells in the mouse, there are clearly more On-types compared to Off (5 Off, 8 On), and the same seems to be true for RGCs (under most stimulus conditions). In fact, if we divide the mouse IPL into 5 layers, as is traditionally done, 2 layers are Off, but 3 are On. But then, the 5th layer is mainly occupied by rod bipolar cells, so if we take that out, we are back at a balanced On versus Off anatomy. In parallel, mammalian Off circuits are somewhat more permissive to many types of stimuli compared to On, meaning that in the end, and depending on the experiment, the code on the optic nerve ends up being more balanced than suggested by retinal anatomy. Primates are even more balanced, an effect that to a large extent is driven by the midgets and parasols which come in both polarities at approximately equal ratios.

Of course, imbalances do exist, perhaps most fundamentally because vertebrate photoreceptors are inherently Off, and On and Off circuits thus cannot be built based on the same synaptic strategy. However, most of these

imbalances are small when contrasted with what we observe in non-mammalian species. In chicken it seems the numerical imbalance is in the order of 5:1 in favour of On, and these correlate with corresponding differences in speed which are not observed in mammals to nearly the same degree. Moreover, On-cells seem to spike more than Off overall, which inflates the imbalance. In zebrafish, depending on the retinal region (for example in their acute zone), the On:Off imbalance is even higher (e.g. Zhou et al. 2020 Curr Biol) – easily exceeding 10:1 in favour of On when looking at their acute zone.

Together, we therefore think that the best description of mammalian encoding of light- and dark contrasts, or of fast and slow contrasts, remains that they are approximately balanced, which by and large has also been the leading narrative of efficient coding theory when applied to mammalian vision. A mouse that readily moves between the open field, the grass, different types of forests, and a burrow, will experience a plethora of natural scene statistics, and therefore the most flexible strategy to deal with this is (probably) to set up the retina to be able to deal approximately equally well with all these contrasts. Of course, a chicken will also need to deal with a similar diversity of contrast statistics, but it seems - based on our first look at the problem and under the caveat of sampling biases as acknowledged above - that they deal with it in a different way.

6. Figure 1 and elsewhere: the temporally accelerating (chirp) stimulus waveform (panel O, bottom) shows distinct aliasing of the stimulus, with basically ON-OFF flicker just prior to 5s, and near-constant ON around 20(?) Does this accurately depict how the stimulus waveform was mapped onto the stimulus monitor/light source? Please address this in figure legend or Methods, and explain how this impacted the stimulus present at the photoreceptors.

Thank you for pointing this out. This is a rendering artifact that resulted from spatial down sampling of the figures when embedding them into the pdf. We now provide full resolution figures where this issue does not appear.

The stimulator ran at 60 Hz, so we opted to probe the chirp up to the Nyquist of 30 Hz. Of course, towards this upper limit there were some small imbalances in the displayed waveform, but we think that these are acceptable in view of the technical limitation in the way that the stimulus was projected, and the fact that most cells clipped long before reaching this limit anyway.

7. Line 151: references non-myelinated axons. Aren't all axons in the retina non-myelinated? My understanding is that myelination begins at the optic nerve head.

Yes, this seems to be the case in mammals. However, in non-mammalian retinas, myelination already within the retina is common. For the specific case of birds, this has been further explored in a recent study: Block et al. 2022 bioRxiv (cited).

8. Line 243: It is shown here that 19.7% of recorded cells were ON, *transient*, non-opponent. The ON sustained fraction is much smaller, just 6.6%. How does this rhyme with the statement on line 229 that 'ON-dominated clusters were sustained'? This is an important discrepancy, as the entire thesis of the manuscript is based on ON = sustained.

Thank you, yes this is a potentially confusing point: What we meant to convey is that even though some On-responses are transient, they are still less transient than the Off responses. This issue is complicated by the fact that many transient On cells do not strongly respond to the "white" steps (Figure 2A), which meant that to split the sustained ones from the more transient ones, we had to separately look at the step responses to the colour flashes. This data is shown in Figure S2A.

Of course, the transience of a flash response is only one way to look at the "speed" of a cell. Other ways might include looking at the kernels or the chirp responses, as summarised in Figure 3. We believe that this data shows quite clearly that on balance, Off-circuits (which includes the Off cluster plus the Off-aspects of the OnOff clusters) are "quicker" than the corresponding On-circuits.

To resolve this confusion, we have rephrased the noted sentence (229) as follows:

Second, polarity and kinetics were linked: Off-dominated clusters were transient, while On-dominated clusters were generally more sustained.

8. Figure 2b: (1) consider making explicit here or in the text what is negative transience. If positive if transient, is zero sustained, and negative a temporally increasing response?

Yes. We now explicitly explain this in the legend, as suggested:

"A positive transience index denotes a transient cell, 0 sustained, and negative a temporally increasing response, as evaluated by comparing the peak response in two time-windows following the step transition: 80-160 ms and 240 – 2,000 ms (Methods)."

(2) if orange symbols are transient and gray symbols sustained, then how can there be gray clusters and orange clusters at the same (negative) transience level?

This is related to the point above, that the transient On cells did not strongly respond to the "white" flash, such that the few spikes that they generate under this condition are in fact weakly facilitating for the chosen time window (cf. Figure 2A). However, when probed with "colour" flashes they are consistently transient (Figure S2A – compare also e.g. Figure 2a C18 white vs colour responses). This "complication" is explained in the results (original line 233ff) and the corresponding legend.

9. This is a comment: Fig2D, E, F – nice that data sets from three different vertebrate species are shown here for comparison.

Thank you!

10. Line 269: typo, as Figure *2*a.

Thanks, fixed!

11. Line 432-433: It is counterintuitive that drive from a single opsin as proposed here for OFF cells would give broad, achromatic tuning. Would their tuning not reflect that of the opsin? Perhaps briefly explain here how this works.

Yes, the RGC's tuning should reflect that of the opsin, however the opsin is broad. This is for a combination of reasons. First, long wavelength opsins, when plotted in the traditional wavelength space, are broader than short wavelength opsins (opsins are approximately "width matched" when looking at energy, but this maps logarithmically onto wavelength). Second, as photoreceptors transform photon flux into synaptic release onto postsynaptic circuits, we expect an approximate log-transform of the signal. Consequently, the effective opsin spectrum that we expect to drive bipolar cells is a log-transformed one compared to the traditional Govardovskii templates that describes the opsin itself.

Taking these factors together, we arrive at the "effective" opsin templates shown in the figures, where the red-one more than any others covers a substantial fraction of visible light. If this was a shallow water aquatic species, like zebrafish, the thus transformed red-opsin template would provide a very good match to the spectral availability of sunlight (e.g. see Bartel et al. 2021 Curr Biol). Of course, the chicken is terrestrial, but because vision first evolved in the well-illuminated shallows, and most vertebrates have retained this original red-cone+LWS opsin, also terrestrial LWS cones are generally considered the achromatic channel (even if technically they do not capture some of the now available short-wavelength light which is lost in the water due to spectral filtering). For example, in primates, we usually consider the two LWS-cone variants as the achromatic channel, where possible added input from SWS1 cones (which are in the great minority) is not essential for much of greyscale vision.

12. Line 672: $ra \cdot n \cdot ge$.

Fixed, thank you.

REVIEWERS' COMMENTS

Reviewer #1 (Remarks to the Author):

The authors have satisfactorily responded to all my questions and made the necessary changes to the manuscript. I would like to congratulate the authors on their very nice work!

Reviewer #2 (Remarks to the Author):

The authors have made a large effort to improve the manuscript and clarify issues pointed out by all three reviewers. This is a comprehensive and detailed study of the functionality of RGC in the chicken retina. I think the manuscript can be published without further revisions.

Reviewer #3 (Remarks to the Author):

I appreciate the authors' responses to my comments in their rebuttal, and the related changes made in the manuscript. The changes and added analyses are solid and satisfactory, and I have no further comments. I think this is very interesting work for the field.